# Genetic basis of sRNA quantitative variation analyzed using an experimental population derived from an elite rice hybrid

**Jia Wang**[1,2†]**, Wen Yao**[1,2†]**, Dan Zhu**[1,2]**, Weibo Xie**[1,2]**, Qifa Zhang**[1,2]*****

[1]National Key Laboratory of Crop Genetic Improvement, Huazhong Agricultural University, Wuhan, China; [2]National Center of Plant Gene Research, Huazhong Agricultural University, Wuhan, China

**Abstract** We performed a genetic analysis of sRNA abundance in flag leaf from an immortalized $F_2$ (IMF2) population in rice. We identified 53,613,739 unique sRNAs and 165,797 sRNA expression traits (s-traits). A total of 66,649 s-traits mapped 40,049 *local*-sQTLs and 30,809 *distant*-sQTLs. By defining 80,362 sRNA clusters, 22,263 sRNA cluster QTLs (scQTLs) were recovered for 20,249 of all the 50,139 sRNA cluster expression traits (sc-traits). The expression levels for most of s-traits from the same genes or the same sRNA clusters were slightly positively correlated. While genetic co-regulation between sRNAs from the same mother genes and between sRNAs and their mother genes was observed for a portion of the sRNAs, most of the sRNAs and their mother genes showed little co-regulation. Some sRNA biogenesis genes were located in *distant*-sQTL hotspots and showed correspondence with specific length classes of sRNAs suggesting their important roles in the regulation and biogenesis of the sRNAs.

*For correspondence: qifazh@
mail.hzau.edu.cn

†These authors contributed
equally to this work

Reviewing editor: Justin
Borevitz, Australian National
University, Australia

## Introduction

Small RNAs (sRNAs) are non-coding RNAs mainly 18–30 nt in length that regulate a wide range of biological processes in eukaryotic organisms (*Carthew and Sontheimer, 2009*; *Axtell, 2013*). According to their origin, sRNAs can be grouped into two major types: hpRNAs that are derived from single-stranded precursors with a hairpin structure (such as microRNAs [miRNAs]) and short interfering RNA (siRNAs) that are derived from double-stranded RNA precursors such as heterochromatic small interfering RNAs (hc-siRNA) and *trans*-acting siRNAs (ta-siRNA). There has been an explosion of interest in recent years in studies of miRNAs and siRNAs on their identification, biogenesis, and functioning in diverse biological processes. In plants, sRNAs function in regulating growth, development (*Juarez et al., 2004*; *Zhu and Helliwell, 2011*), architecture (*Jiao et al., 2010*; *Miura et al., 2010*), yield (*Zhang et al., 2013*), and response to biotic and abiotic stresses (*Lu et al., 2008*; *Shukla et al., 2008*). Such regulations are usually achieved by mediating endogenous mRNA cleavage and decay, DNA methylation of source and target loci, and chromatin modification and transcriptional silencing (*Arikit et al., 2013*).

Although the sRNAs differ in length, sequences, and functions, the pathways for their biogenesis and functioning from precursor transcription, processing, maturation, and action are relatively conserved, which involve the activities of a number of enzymes including RNA polymerase II (Pol II), RNA-dependent RNA polymerases (RDRs), Dicer-like proteins (DCLs), and Argonautes (AGOs) (*Chen, 2009*; *Ghildiyal and Zamore, 2009*). It is also known that the abundance of sRNA species can be highly variable among individuals within a species, and the possible regulatory role of such

**eLife digest** Genes within the DNA of a plant or animal contain instructions to make molecules called RNAs. Some RNA molecules can be decoded to make proteins, whereas others have different roles. A single gene often contains the instructions to make both protein-coding RNAs and non-coding RNAs.

Molecules called small RNAs (or sRNAs) do not code for proteins. Instead, sRNAs can control protein-coding RNA molecules or chemically alter the DNA itself; this allows them to perform many different roles in living organisms. In plants, for example, these molecules affect how the plant grows, the shapes and structures it forms, and how likely it is to survive challenges such as drought and diseases. Often different plants of the same species have different amounts of sRNAs, but the reasons for this remain unclear.

Now, Wang, Yao et al. have made use of a technique called 'expression quantitative locus' analysis to look at how sRNAs in rice plants are controlled by additional information encoded within DNA. The analysis identified over 53 million sRNA molecules from a population of rice plants. Many of these sRNAs varied in their abundance between different plants within the population. Wang, Yao et al. also found many thousands of individual instructions within the DNA of the rice that can either increase or reduce the abundance of their associated sRNA.

Some of the abundant sRNAs were influenced by instructions within their own genes; some were influenced by instructions from other genes; and some were influenced by both. Wang, Yao et al. also found that the control of protein-coding RNAs was not necessarily related to the control of sRNAs encoded by the same gene. Further work is now needed to identify which specific DNA sequences regulate the abundance of sRNA molecules in plants and other organisms.

quantitative difference has been assumed (*He et al., 2010*; *Groszmann et al., 2011b*). It is not known whether the quantitative variation of sRNA species between genotypes is related to the biological machinery, as quantitative variation of sRNAs has not been assayed at the population level and their genetic control has yet to be elucidated.

The recently developed expression quantitative trait locus (eQTL) analysis has provided an approach for determining the genetic control of the expression level of a gene, including *cis*- and *trans*-eQTLs, as well as epistatic effects (*Becker et al., 2012*). This approach can also be applied to the genetic analysis of quantitative variation of sRNAs by regarding the abundance of the sRNAs in the population as quantitative traits. Once the QTLs are identified, subsequent studies can be pursued very much the same way as the analysis of genes and regulatory networks underpinning phenotypic QTLs (*Xing and Zhang, 2010*).

There have been several studies focusing on the genetic regulation of known and validated miRNAs and small nucleolar RNAs in specific tissues/cells from samples of the human population (*Borel et al., 2011*; *Gamazon et al., 2012*; *Parts et al., 2012*; *Civelek et al., 2013*; *Jin and Lee, 2013*; *Siddle et al., 2014*). These studies detected a number of *cis*- and *trans*-miQTLs that could also influence the expression of the mRNA targets, which may be associated with phenotype difference.

Here, we performed a whole genome QTL analysis of the entire sRNA kingdom consisted of 18-nt to 26-nt sRNAs from flag leaf of rice using an experimental genetic population. The analysis revealed features of the genetic controls of sRNA abundance showing both commonality and distinction with their precursor transcripts. It was also shown that the abundance of sRNAs is probably related to proteins constituting the machinery for sRNA biogenesis and functioning.

## Results

### Patterns and distributions of sRNAs

The genetic materials consisted of 98 hybrids obtained by paired crosses of 196 recombinant inbred lines (RILs) derived by single seed descent from a cross between Zhenshan 97 and Minghui 63, the parents of Shanyou 63 that was the most widely cultivated rice hybrid in the 1980s and 1990s and still used with reduced area in recent years in China. Because the genetic composition of these crosses resembled individuals in an $F_2$ population, they were referred to as an immortalized $F_2$ (IMF2)

population (*Hua et al., 2002*, *2003*), and each hybrid in the population was hereafter referred to as an IMF2 for ease of description. An sRNA library was constructed using RNA extracted from flag leaf at the day of full expansion for each IMF2, and two biological replicates were obtained for each of the two parental lines and their $F_1$ hybrid, producing a total of 104 libraries.

The read size of the raw sequencing data obtained using Illumina Hiseq2000 varied from 3 nt to 44 nt (*Figure 1—figure supplement 1*). Sequences of 18–26 nt in length that appeared to be the more abundant than others were kept for the analyses after filtering out low-quality reads and eliminating ones matching tRNAs, rRNAs, snRNAs, and snoRNAs (*Figure 1—figure supplement 2*). The numbers of resulting reads varied from 14.52 million to 27.73 million per library (2.08 billion in total) with non-redundant reads ranging from 3.00 million to 6.86 million per library (0.52 billion in total) (Supplementary file 1 in Dryad [*Wang et al., 2015*], *Table 1*, *Figure 1—figure supplement 2*). The 24 nt sRNAs were the most numerous in both redundant and distinct reads (*Figure 1—figure supplement 2*).

We mapped the reads to the SNP-replaced reference genomes of the parents ('Materials and methods'), with unique location allowing no mismatch. The reads could be divided into four categories: (1) approximately 82.28% of the sRNAs had identical sequences between parents, (2) 0.71% of the sRNAs had SNPs between the two parents, (3) 8.35% were only mapped to the Zhenshan 97 genome, and (4) 8.66% were specifically mapped to the Minghui 63 genome (Supplementary file 2 in Dryad [*Wang et al., 2015*]). A total of 53,613,739 unique sRNA sequences including ones with SNPs between the parents were identified by combining all the reads from 104 libraries (*Table 1*). Approximately 84.7% of sRNAs were found in no more than 5 IMF2s, while only 0.13% were present in all 98 IMF2s (*Figure 1A*).

The distributions of the sRNAs in different portions of the genome were not random. Approximately 51.1% of sRNAs originated from the 2-kb upstream and the genic regions of non-transposon genes (*Figure 1B*). In addition, sRNAs occurred in high frequencies near the transcription start sites compared to other regions of the promoters (*Figure 1—figure supplement 3*). About 34.4% of sRNAs originated from intergenic regions, which account for no more than 20% of the whole genome length (*Figure 1B*). About 43.9% of sRNAs were 24 nt, while 12.1% were 21 nt (*Figure 1C*).

Different species of sRNAs differed in their origins of genomic regions. The genic regions of non-transposon genes held the highest number of 21 nt sRNAs, while 24 nt sRNAs were most common in the intergenic regions (*Figure 1D*). sRNAs of 21 nt in genic regions were mostly derived from the exon of non-transposon genes, while 24 nt sRNAs in genic regions were mainly from the intron of non-transposon genes (*Figure 1E*). The 24-nt sRNAs mainly consist of endogenous heterochromatic siRNAs (*Axtell, 2013*) and tend to be produced from repeats and transposable elements as well as intergenic regions (*Chen, 2009*). The 24-nt sRNAs are components of the epigenome that target the homologous genomic regions for de novo DNA methylation through RNA-directed DNA methylation to maintain genome stability by transcriptional gene silencing (*Groszmann et al., 2011a*, *2013*; *Pooggin, 2013*).

Next, the abundance of each sRNA in a library was normalized to number of reads per millions (RPM) ('Materials and methods') to quantify the sRNA levels in each library, which were subject to analyses and comparisons. An sRNA with RPM value ≥0.6 was regarded as expressed, and an sRNA identified as expressed in more than 25 of the 98 IMF2s was regarded as an sRNA expression trait (s-trait). In this way, a total of 165,797 s-traits were recovered (Supplementary file 3 in Dryad

**Table 1**. Average number of reads obtained for the 104 libraries from flag leaves of the IMF2 population and the parental lines

|  | Quality filtering* | Size filtering† | NcRNA filtering‡ | Genome mapping§ | Total# |
|---|---|---|---|---|---|
| Redundant reads | 22,864,067 | 22,318,928 | 19,968,566 | 9,816,933 | 1,020,961,021 |
| Distinct reads | 5,106,654 | 5,035,073 | 4,986,184 | 2,499,233 | 53,613,739 |

*Reads after filtering out low-quality reads.
†Reads of 18–26 nt in length.
‡Reads after eliminating ones matching tRNAs, rRNAs, snRNAs, and snoRNAs.
§Reads mapped to the SNP-replaced reference genomes of the parents with unique locations allowing no mismatch.
#The total reads of 104 libraries used to identify s-traits.

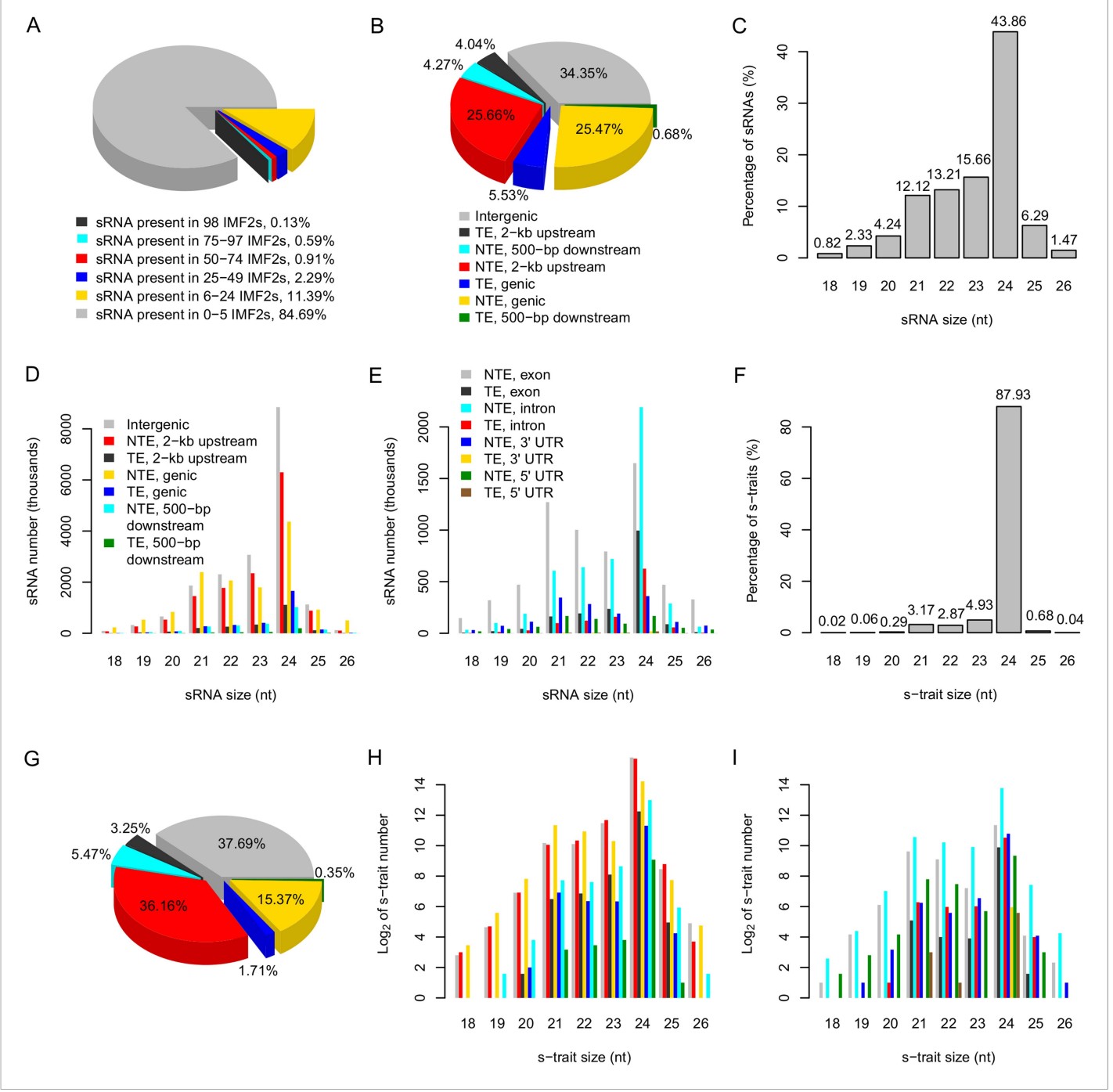

**Figure 1**. The distribution of sRNAs and s-traits in different genomic regions across the IMF2 population. (**A**) The distribution of sRNAs across the IMF2 population. (**B**) The distribution of sRNAs aligned to the genic region, 2-kb upstream, and 500-bp downstream of annotated genes, as well as intergenic regions. TE: transposons; NTE: non-transposon genes. (**C**) The percentages of sRNAs of different sizes. (**D**) The distribution of sRNAs of different sizes in different genomic regions. (**E**) The distribution of sRNAs of different sizes in different portions of genic regions. (**F**) The percentages of s-traits of different sizes. (**G**) The distribution of s-traits aligned to the genic region, 2-kb upstream, and 500-bp downstream of annotated genes, as well as intergenic regions. The legend is the same as in (**B**). (**H**) The distribution of s-traits of different size in different genomic regions. The legend is the same as in (**D**). (**I**) The distribution of s-traits of different sizes in different portions of genic regions. The legend is the same as in (**E**).

The following figure supplements are available for figure 1:

**Figure supplement 1**. The number of sRNA reads of different lengths in the raw sequencing data of the IMF2 population.

*Figure 1. continued on next page*

*Figure 1. Continued*

**Figure supplement 2**. Processing of raw sRNA reads to genome-mapped reads.

**Figure supplement 3**. Distribution of sRNAs in the upstream and downstream of genic regions.

**Figure supplement 4**. The distribution of coefficients of variation (CV) for 165,797 s-traits.

**Figure supplement 5**. The distributions of the expression values across all the IMF2s for 9 randomly selected s-traits.

[*Wang et al., 2015*]). A total of 87.9% of s-traits were 24 nt (*Figure 1F*). The s-traits mainly originated from the intergenic regions, 2-kb upstream, and genic regions of non-transposon genes (*Figure 1G*). Although the 2-kb upstream and the genic regions of non-transposon genes contained almost equal number of sRNAs (*Figure 1B*), the number of s-traits derived from the 2-kb upstream of non-transposon genes was more than twice of the number of s-traits in genic regions of non-transposon genes (*Figure 1G*). The distribution of different species s-traits in different genomic regions was similar to that of sRNAs (*Figure 1D,H*). Although sRNAs of various lengths from genic regions were mostly from exons of non-transposon genes (*Figure 1E*), the s-traits of different sizes were mainly derived from introns of non-transposon genes (*Figure 1I*). The coefficients of variation of the 165,797 s-traits (*Figure 1—figure supplement 4*) showed a very wide range of distribution of the sRNA levels in the population. We selected 9 s-traits randomly from all the 165,797 s-traits, and the distributions of the expression values for most of these s-traits across the IMF2 population were more or less normal (*Figure 1—figure supplement 5*).

## Expression correlations between sRNAs that originated from the same genes

To test whether s-traits originated from the same gene were transcribed together, the correlation coefficients between the expression values of s-traits originated from the same gene were calculated. A total of 3,739,873 correlation coefficients involving 15,078 genes and 96,914 s-traits were obtained, 2,593,503 (69.3%) and 507,528 (13.6%) of which were contributed by 2278 s-traits from genes LOC_Os03g01360 and 1008 s-traits from LOC_Os07g01240, respectively (*Figure 2*). The correlation coefficients naturally fell into three classes: strong negative correlations, no correlation, and strong positive correlations with obvious dividing points at −0.3 and 0.3 (*Figure 2A*), which were far above 0.25, the threshold for statistical significance at p < 0.05 determined by simulation data. While 85.8% of these strong correlations (positive or negative) were contributed by s-traits originated from LOC_Os03g01360 (*Figure 2B*), most of s-traits originated from gene LOC_Os07g01240 showed low (or no) correlations with each other (*Figure 2C*). Although the expression values of s-traits originated from the same regions of the remaining 15,076 genes in the whole genome were mostly slightly positively correlated, the correlation coefficients were significantly distinct from that of the simulation of random data (Mann–Whitney U test, p-value <2.2e-16, *Figure 2D*), indicating various degrees of correlations. We found that the correlations were affected by the sizes of s-traits (*Figure 2—figure supplement 1*). Expression correlations between s-traits of the same size were stronger than that between s-traits of different sizes. In particular, the expression correlations between 21-nt sRNAs and 25-nt sRNAs from the same genes were very weak. The correlations between overlapped s-traits were slightly stronger than that between s-traits not overlapping with each other (*Figure 2—figure supplement 2A,B*). The correlations between s-traits from the same transposons were slightly different from that of s-traits from the same non-transposons genes (*Figure 2—figure supplement 2C,D*). Most of the s-traits originating from LOC_Os03g01360 were significantly correlated (positive or negative) with each other (*Figure 2—figure supplement 3*), and the correlations between different s-traits were mainly dependent on their genomic positions rather than their sizes, which were mostly 21, 22, and 24 nt, accounting for 40.2%, 34.4%, and 19.4% of the 2278 s-traits. This gene could be separated as three parts, the 2 kb-upstream, 5′ UTR, the first intron and the first coding DNA sequence (CDS) as the first part, the most of the second intron comprising the second part, and the second CDS and part of the second intron as the third part (*Figure 2—figure supplement 3*). s-traits

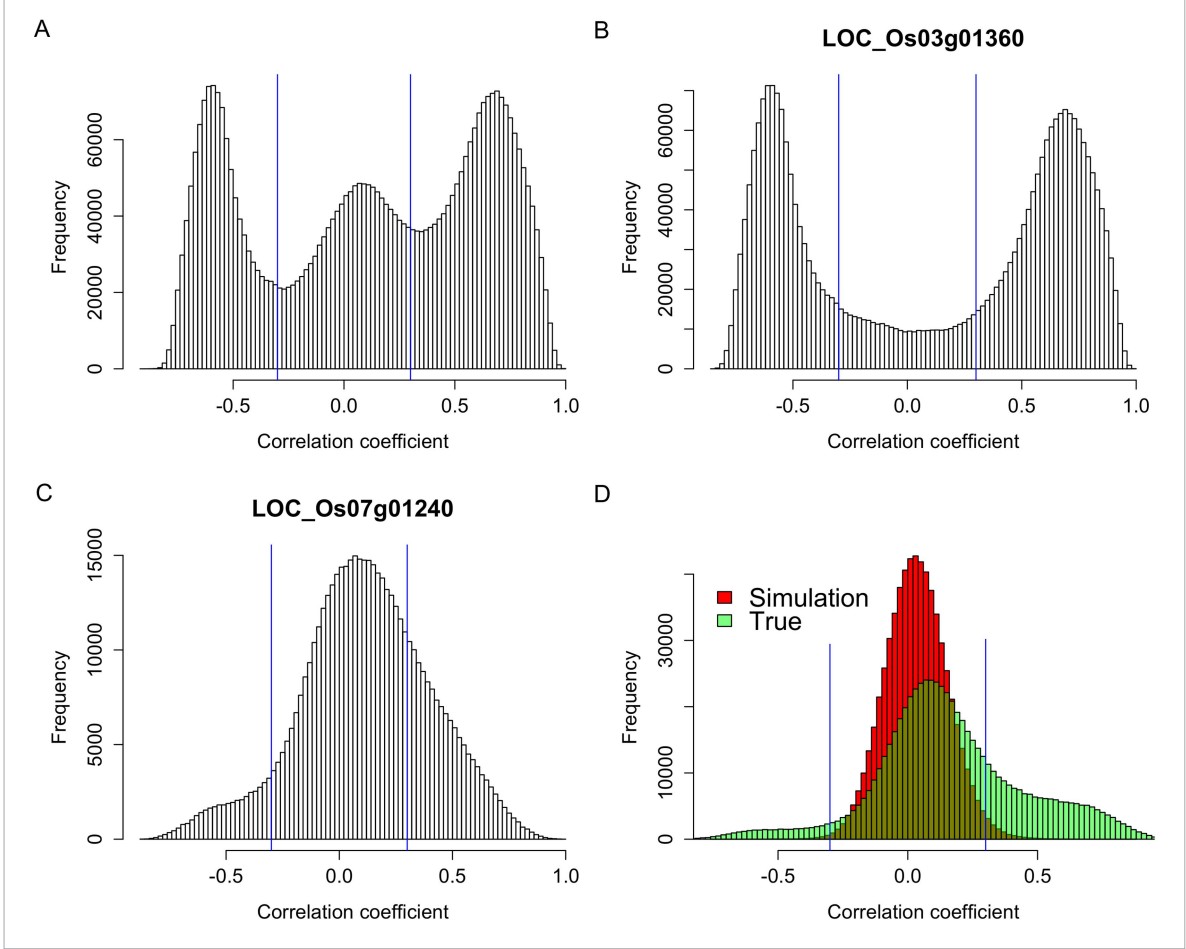

**Figure 2**. Expression correlations between s-traits originating from the same genes. (**A**) The distribution of expression correlations between s-traits originating from the same genes. (**B**) The distribution of expression correlations between s-traits originating from LOC_Os03g01360. (**C**) The distribution of expression correlations between s-traits originating from LOC_Os07g01240. (**D**) Comparison of the distribution of expression correlations for the simulation data and the true data excluding s-traits from LOC_Os03g01360 and LOC_Os07g01240.

The following figure supplements are available for figure 2:

**Figure supplement 1**. Expression correlations between s-traits of different sizes originating from the same genes.

**Figure supplement 2**. Expression correlations between s-traits originating from different regions of the same mother genes.

**Figure supplement 3**. The expression correlations between s-traits originating from LOC_Os03g01360.

**Figure supplement 4**. Distribution of sRNA cluster sizes.

**Figure supplement 5**. Expression correlations between s-traits originating from the same sRNA clusters.

within each part were positively correlated with each other, while s-traits from the first part were mostly negatively correlated with those from the other two parts. The correlations between the expression levels of s-traits from the second parts and s-traits from the third parts were mostly positive. Thus, although very strong positive and negative correlations were found between sRNAs from the same genic regions of specific genes, this was not a common phenomenon. Taken together, this correlation analysis indicated that most of the sRNAs derived from the same genes were correlated with each other to various extents, implying that nearby sRNAs were possibly jointly

transcribed. Consecutively overlapped sRNAs were found from introns of specific genes and intergenic regions, indicating that sRNAs were not necessarily jointly transcribed from the genes.

## Expression correlations between sRNAs that originated from the same sRNA cluster

As many sRNAs may be derived from a single primary transcript, we defined sRNA clusters to quantify sRNA expression (*Castel and Martienssen, 2013*), based on our data of all 104 libraries. An sRNA island was defined as a genomic region composed of consecutive genomic positions matched by at least 30× sRNA read coverage. Nearby sRNA islands resided within 1000 nt from each other were merged and regarded as the same sRNA cluster. As a result, 80,362 sRNA clusters were recovered with an average size of 1927 bp (ranging from 51 bp to 47,021 bp) (*Figure 2—figure supplement 4*, Supplementary file 4 in Dryad [*Wang et al., 2015*]). A total of 46,293 (57.6%) sRNA clusters overlapped with genic regions, 23,143 (28.8%) of which were entirely derived from genic regions. In addition, the distances between another 19,164 (23.8%) sRNA clusters and the genic regions were smaller than 1 kb.

The abundance of each sRNA cluster was normalized using DEseq ('Materials and methods') (*Anders and Huber, 2010*). An sRNA cluster with normalized expression value ≥6 was regarded as expressed, and an sRNA cluster expressed in more than 25 of all the 98 IMF2s was regarded as an sRNA cluster expression trait (sc-trait). A total of 50,139 (62.4%) sRNA clusters were determined as sc-traits, 34,762 of which were expressed in all of the 98 IMF2s.

We next calculated the correlation coefficients between the expression values of s-traits originating from the same sRNA clusters to infer whether they were transcribed together. A total of 4,282,587 correlation coefficients involving 21,571 sRNA clusters and 157,994 s-traits were obtained, 72.5% of which were contributed by s-traits from clusters chr03-261504-273325 and chr07-136331-141317 (*Figure 2—figure supplement 5*). These two sRNA clusters corresponded to the two genes, LOC_Os03g01360 and LOC_Os07g01240, mentioned above. Of all the correlations, 2,776,296 were relatively strong (above 0.3 or below −0.3 based on a threshold of 0.25 determined by simulation data with p <0.05). However, 81.5% of these strong correlations were contributed by the 2280 s-traits from cluster chr03-261504-273325 (*Figure 2—figure supplement 5B*). A total of 1008 s-traits originated from cluster chr07-136331-141317, most of which were only slightly correlated with each other (*Figure 2—figure supplement 5C*). The distribution of correlation coefficients of s-traits from the same sRNA clusters was quite similar to that of s-traits from the same genes (*Figure 2—figure supplement 5D*). Although the s-traits originating from the same region of the remaining 215,69 sRNA clusters in the whole genome were only slightly positively correlated, the correlation coefficients were significantly different from that of simulated random data (Mann–Whitney U test, p-value <2.2e-16, *Figure 2—figure supplement 5E*).

Thus, again very strong positive and negative expression correlations were only observed for sRNAs from certain sRNA clusters, while most sRNAs originating from the same sRNA clusters were only slightly positively correlated with each other.

## Expression correlations between sRNAs and their mother genes

mRNA sequencing was conducted using the same samples as in the sRNA sequencing with one biological replicate for the IMF2 population and two biological replicates for each of the two parental lines and their $F_1$ hybrid, producing a total of 104 libraries. After removal of low-quality sequencing data, reads were mapped to the Nipponbare reference genome using TopHat (*Figure 3—figure supplement 1*, Supplementary file 5 in Dryad [*Wang et al., 2015*]) (*Trapnell et al., 2009*). Most of the reads were mapped to the CDS and UTR regions of non-transposon genes (*Figure 3—figure supplement 2*).

The abundance of each mRNA in a library was normalized to the number of fragments per kilobase of transcript per million mapped fragments (FPKM) using Cufflinks (*Trapnell et al., 2010*) to quantify the expression levels of mRNAs in each library. An mRNA with FPKM value ≥1 was regarded as expressed; 35,819 (54.2%) mRNAs were expressed in none of the 98 IMF2s, while 16,849 (25.5%) were expressed in all the 98 IMF2s. An mRNA identified as expressed in more than 25 of the 98 IMF2s was regarded as an e-trait. In this way, a total of 24,987 e-traits were obtained from a total of 66,123 mRNAs in the whole genome.

For ease of description, we hereafter refer to genic sequences including both exons and introns that encoded the s-traits as 'mother genes'. To assess whether sRNAs and their mother genes were

transcribed together, the correlation coefficients of the expression values of s-traits and their mother genes were calculated. A total of 77,640 correlation coefficients were obtained, involving 9968 mRNAs and 56,185 s-traits. Among all these correlations, 13,749 (17.7%) were strong correlations (above 0.3 or below −0.3, based on a threshold of 0.23 determined by simulation data with p < 0.05). However, 45.6% of these strong correlations were again contributed by gene LOC_Os03g01360 (*Figure 3A*). Correlations involving LOC_Os07g01240 were mostly weak (*Figure 3B*). The correlation coefficients involving rest of the genes in the whole genome were centered around zero (*Figure 3C*).

In-depth analysis of s-traits from gene LOC_Os03g01360 revealed that the expression values of s-traits from the first part of LOC_Os03g01360 were positively correlated with two transcripts, LOC_Os03g01360.1 and LOC_Os03g01360.4, and negatively correlated with the other LOC_Os03g01360.2 (*Figure 3D*). On the other hand, the expression values of most s-traits from the third part of LOC_Os03g01360 were negatively correlated with LOC_Os03g01360.1 and LOC_Os03g01360.4 but positively correlated with LOC_Os03g01360.2 (*Figure 3D*). Whereas both negative and positive expression correlations were found between s-traits from the second part of LOC_Os03g01360 and these three transcripts (*Figure 3D*).

Thus, although strong expression correlations, either positive or negative, were observed between specific sRNAs and their mother gene, the expression levels of most sRNAs were not correlated with their mother genes, suggesting independent transcription of sRNAs and mRNAs.

## QTL analysis of s-traits, sc-traits, and e-traits using the IMF2 population

QTL analysis was performed for the 165,797 s-traits based on the ultrahigh-density SNP map, which contained 1556 recombination events and was composed of 1568 bins with an average size of 238 kb (ranging from 6 kb to 7947 kb) (*Figure 4—figure supplement 1*, *Figure 4—figure supplement 2*, Supplementary file 6 in Dryad [*Wang et al., 2015*], 'Materials and methods') (*Xie et al., 2010*; *Yu et al., 2011*), using composite interval mapping (CIM) in R/qtl with 1000 permutations (*Haley and Knott, 1992*; *Broman and Speed, 2002*; *Manichaikul et al., 2009*). With a false discovery rate (FDR) set at 5%, a total of 70,858 sQTLs were recovered for 66,649 s-traits (Supplementary file 7 in Dryad [*Wang et al., 2015*]). These sQTLs were classified as *local*-sQTLs and *distant*-sQTLs, which were also referred to as *cis*- and *trans*-QTLs, respectively, in previous studies (*Wang et al., 2010*, *2014*), according to their locations relative to the s-traits. A *local*-sQTL indicated the existence of local functional polymorphism(s) that could influence the abundance of the s-trait, and a *distant*-sQTL meant the s-trait expression variation in the population was controlled by regulatory element(s) distant from the s-trait precursor sequence. For defining a *local*-sQTL, we adopted a 1.5 LOD-drop support interval of the corresponding sQTL or no more than 250 kb from the closest marker in a recombination sparse region. Otherwise, it was regarded as a *distant*-sQTL (*Yvert et al., 2003*; *Morley et al., 2004*; *Keurentjes et al., 2007*). Such classification resulted in 40,049 *local*-sQTLs (shown in the diagonal of *Figure 4*) and 30,809 *distant*-sQTLs (off-diagonal of *Figure 4*) (Supplementary file 7 in Dryad [*Wang et al., 2015*]). Expression variations of s-traits explained by *local*-sQTLs were substantially higher than that by *distant*-sQTLs (*Figure 4—figure supplement 3*). The LOD values for 80.4% of the *local*-sQTLs and 34.5% of *distant*-sQTLs were larger than 10.

Next, we performed QTL analysis for 50,139 sc-traits based on the same genetic map as in sQTL analysis using CIM in R/qtl with 1000 permutations. A total of 22,263 scQTLs including 11,476 *local*-scQTLs and 10,787 *distant*-sQTLs were recovered for 20,049 sc-traits utilizing the same definitions of *local*- and *distant*-QTLs (*Figure 4—figure supplement 4*, Supplementary file 7 in Dryad [*Wang et al., 2015*]). Again, the LOD values of *local*-scQTLs and expression variations explained by *local*-scQTLs were much higher than *distant*-scQTLs (*Figure 4—figure supplement 5*).

We also performed QTL analysis for the 24,987 e-traits based on the same genetic map, using the same algorithm and parameters as in the QTL analysis of s-traits and sc-traits. With an FDR set at 5%, a total of 6423 eQTLs were recovered for 6123 e-traits (Supplementary file 7 in Dryad [*Wang et al., 2015*]). These eQTLs were classified as *local*-eQTLs and *distant*-eQTLs according to the same rule applied in the sQTL analysis. As a result, 2964 *local*-eQTLs (shown in the diagonal of *Figure 4—figure supplement 6*) and 3459 *distant*-eQTLs (off-diagonal of *Figure 4—figure supplement 6*) were detected (Supplementary file 7 in Dryad [*Wang et al., 2015*]). Expression variations of mRNAs explained by *local*-eQTLs were significantly higher than that by *distant*-eQTLs (*Figure 4—figure supplement 7*).

*Local*-QTLs regulating the expression of large numbers of traits were observed for s-traits and sc-traits but not for e-traits (*Figure 4*, *Figure 4—figure supplement 4* and *Figure 4—figure supplement 6*),

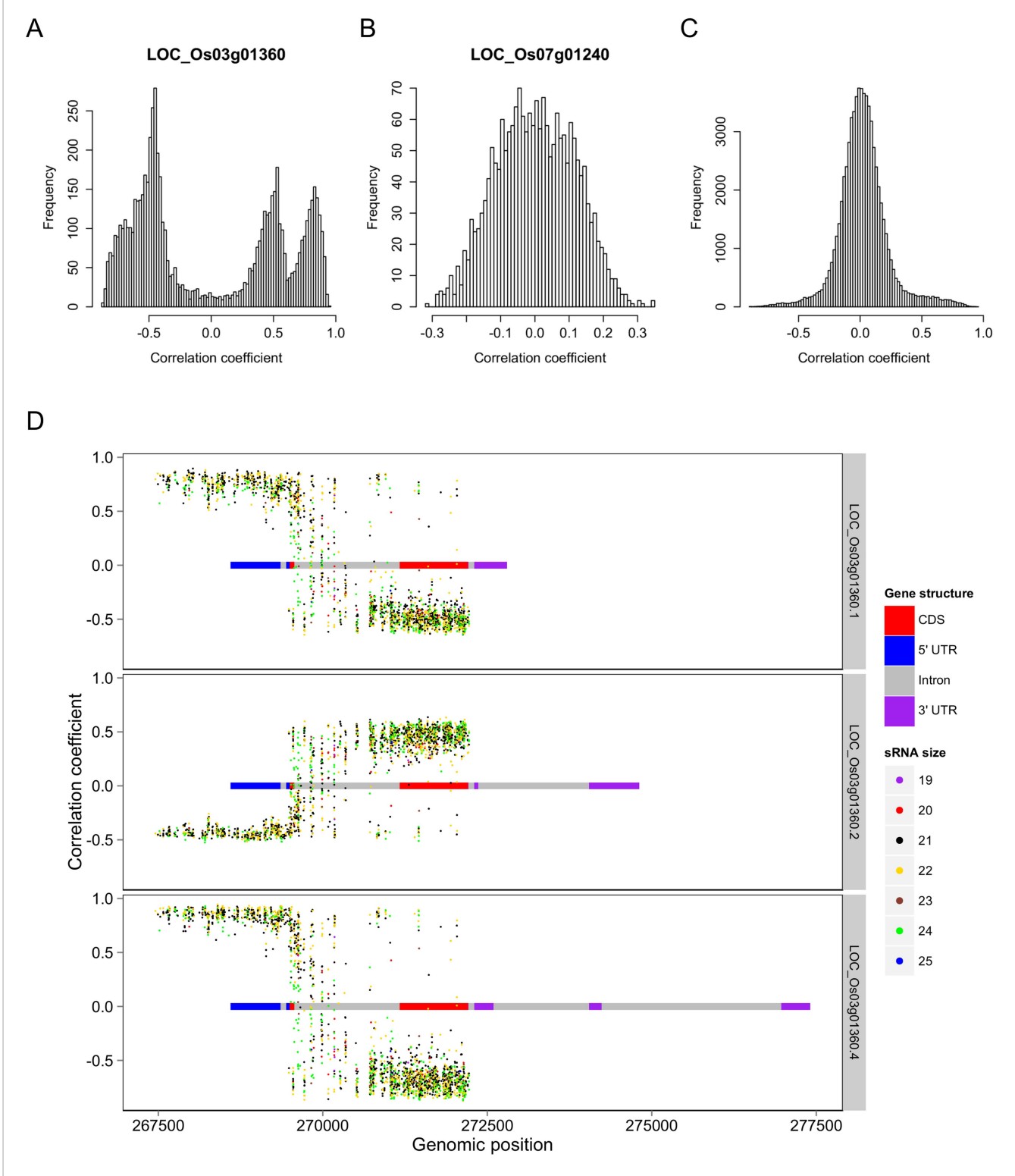

**Figure 3**. Correlations between the s-traits and the transcripts of their mother genes. (**A**) Correlations between s-traits and e-traits derived from LOC_Os03g01360. (**B**) Correlations between s-traits and e-traits derived from LOC_Os07g01240. (**C**) Correlations between s-traits and e-traits originating from the same genes excluding the above two loci. (**D**) Correlations between s-traits and different e-traits derived from LOC_Os03g01360.

The following figure supplements are available for figure 3:

*Figure 3. continued on next page*

Figure 3. Continued

**Figure supplement 1**. Processing of mRNA sequencing data from raw sequencing reads to genome-mapped reads.

**Figure supplement 2**. The distribution of mRNA sequencing reads in different genomic regions.

probably due to the consecutive transcription of sRNAs and sRNA clusters, which was distinct from the transcription of mRNAs. In the extreme case, a total of 1660 s-traits were regulated by a single *local*-sQTL (Bin358) on chromosome 3 (Supplementary file 8 in Dryad [*Wang et al., 2015*]). Each of another five *local*-sQTLs (Bin359, Bin1183, Bin969, Bin286, and Bin166) explained the expression variations of more than 200 s-traits. The *local*-scQTL (Bin1183) with the largest effect regulated the expression of 93 sc-traits. Each of another 29 *local*-scQTLs was responsible for the expression variations of more than 30 sc-traits. In comparison, a total of 24 e-traits were regulated by Bin3, which is the *local*-eQTL explaining the expression variations of the largest number of e-traits. *Distant*-QTLs regulating many traits across all chromosomes were very common for s-traits and sc-traits but relatively rare for e-traits. A total of 15 bins were identified as *distant*-sQTLs, each of which regulated more than 200 s-traits; in one case, 514 s-traits were regulated by a single *distant*-sQTL (Bin1209). We observed three *distant*-scQTLs (Bin634, Bin635, and Bin636), each of which regulated more than 100 sc-traits, and another 21 *distant*-scQTLs, each of which explained the expression variations of more than 30 sc-traits. Whereas in case of mRNAs, only 9 bins were each responsible for the expression of more than 30 e-traits.

We further inspected the QTL results of 22,142 s-traits expressed (RPM >0) in only one of the two parents. In all, 11,194 and 10,948 s-traits were expressed only in the genome of Zhenshan 97 and Minghui 63, respectively. A total of 15,037 QTLs, 11,781 of which were *local*-sQTLs, were obtained for 14,139 of these s-traits. This proportion was significantly higher than the whole genome average level (Fisher's exact test, p value <2.2e-16), implying that the expression variations of these s-traits were mostly controlled by local polymorphisms. Examples of s-traits expressed in only one of the two parents were shown in *Figure 4—figure supplement 8*.

## Identification of QTL hotspots

We next identified bins enriched with traits or QTLs based on the relative densities of the distributions in the genome ('Materials and methods'). 40 bins were identified as hotspots for s-traits (Supplementary file 8 in Dryad [*Wang et al., 2015*], *Figure 5*). A total of 132 and 129 bins were identified as *local*-sQTL and *distant*-sQTL hotspots, respectively (Supplementary file 8 in Dryad [*Wang et al., 2015*], *Figure 5*). 21 bins were both s-trait and *local*-sQTL hotspots, while 64 bins were both *local*-sQTL and *distant*-sQTL hotspots (Supplementary file 8 in Dryad [*Wang et al., 2015*], *Figure 5*); 123 were *local*-scQTL hotspots and 149 *distant*-scQTL hotspots (Supplementary file 8 in Dryad [*Wang et al., 2015*]). In all, 76 bins were *local*-QTL hotspots and 80 bins were *distant*-QTL hotspots for both s-traits and sc-traits. On the other hand, 116 and 126 bins were identified as *local*- and *distant*-eQTL hotspots. The identified hotspots of sQTLs and scQTLs were quite similar but distinct from that of eQTLs (*Figure 5*). Totally, 33 bins were *local*-QTL hotspots and 36 bins were *distant*-QTL hotspots for both s-traits and e-traits. Consecutive *distant*-QTL hotspots in adjacent genomic regions could explain the expression variations of a large number of s-traits, sc-traits, or e-traits (Supplementary file 8 in Dryad [*Wang et al., 2015*]). Four consecutive *distant*-QTL hotspots on chromosome 4, 5, 6, and 9 were identified for both s-traits and sc-traits (*Figure 5A,B*). The genomic positions of consecutive *distant*-eQTLs hotspots were quite distinct from that of sQTL or scQTLs (*Figure 5*). A cluster of the consecutive *distant*-eQTL hotspots (chr04: 1.713 Mb–2.495 Mb) was located close to one end of chromosome 4, while another cluster of consecutive *distant*-QTL hotspots (chr04: 21.997 Mb–23.342 Mb) for both s-traits and sc-traits was about 10 Mb distant from the other end of chromosome 4. A cluster of consecutive *distant*-eQTL hotspots (chr07: 28.058 Mb–28.557 Mb) was located close to one end of chromosome 7, while no apparent consecutive *distant*-QTL hotspots for s-traits and sc-traits were on chromosome 7. A total of 12 bins (Bin1199–Bin1210) on chromosome 9 were identified as consecutive *distant*-QTL hotspots for both s-traits and sc-traits but not for mRNA e-traits. Only the consecutive *distant*-QTL hotspot on chromosome 6 (chr06: 2.847 Mb–4.023 Mb) was responsible for the expression variations of s-traits, sc-traits, and e-traits.

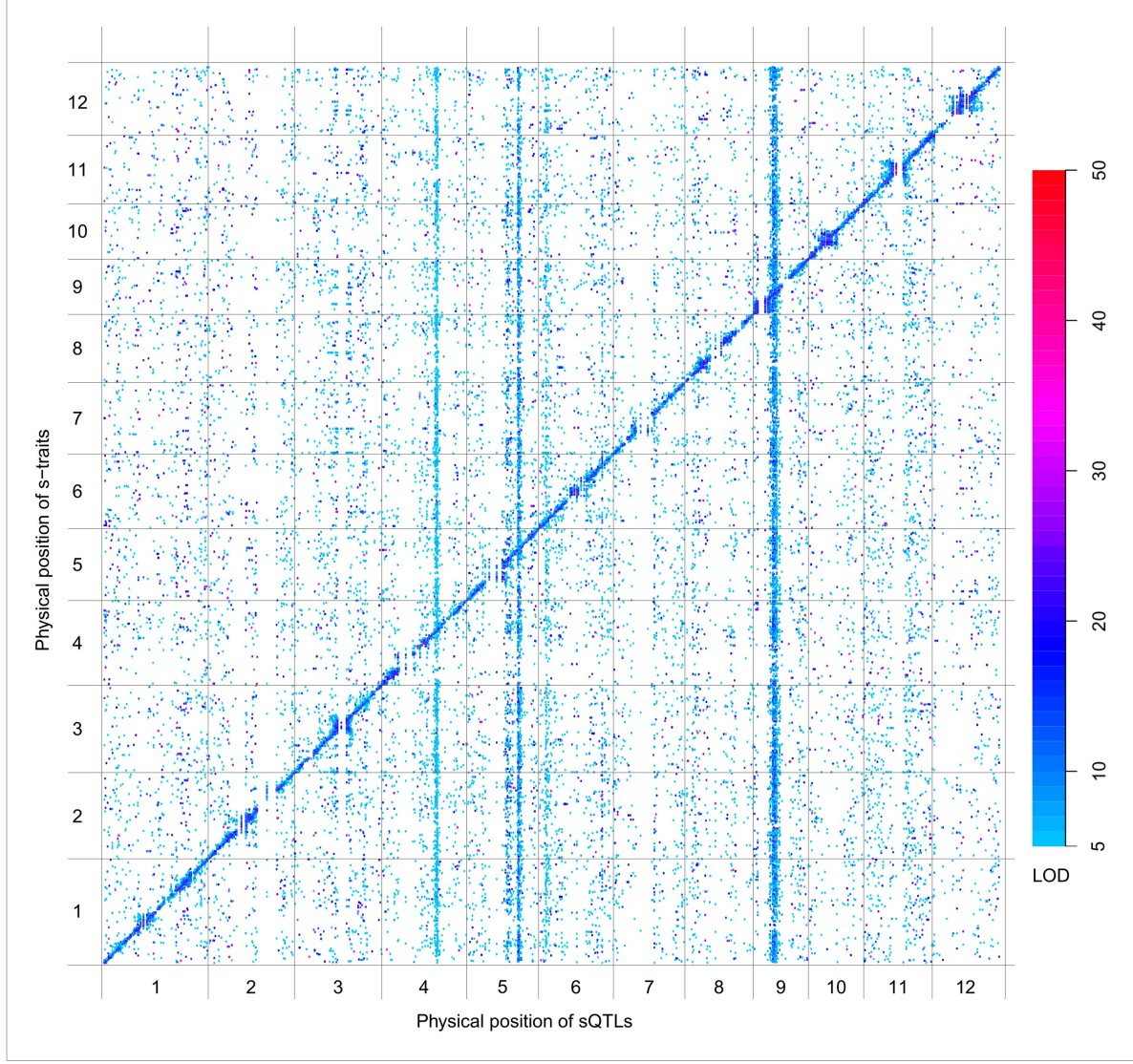

**Figure 4**. sQTLs for the 66,649 s-traits. The color key shows the LOD value. X-axis, the physical position of sQTLs along the genome of 12 chromosomes. Y-axis, the physical position of s-traits. QTLs with LOD value <5 are not included in the presentation.

The following figure supplements are available for figure 4:

**Figure supplement 1**. The distribution of the sizes of 1568 bins of the genetic map.

**Figure supplement 2**. The genotypes of 98 IMF2s across the 1568 bins.

**Figure supplement 3**. The LOD values and expression variations explained by sQTLs ($R^2$) for *local-* and *distant*-sQTLs.

**Figure supplement 4**. scQTLs for the 20,049 sc-traits.

**Figure supplement 5**. The LOD values and expression variations explained by scQTLs ($R^2$) for *local-* and *distant*-scQTLs.

**Figure supplement 6**. eQTLs for the 6123 e-traits.

**Figure supplement 7**. The LOD values and expression variations explained by eQTLs ($R^2$) for *local-* and *distant*-eQTLs.

**Figure supplement 8**. Examples of s-traits expressed in only one of the two parents (Zhenshan 97).

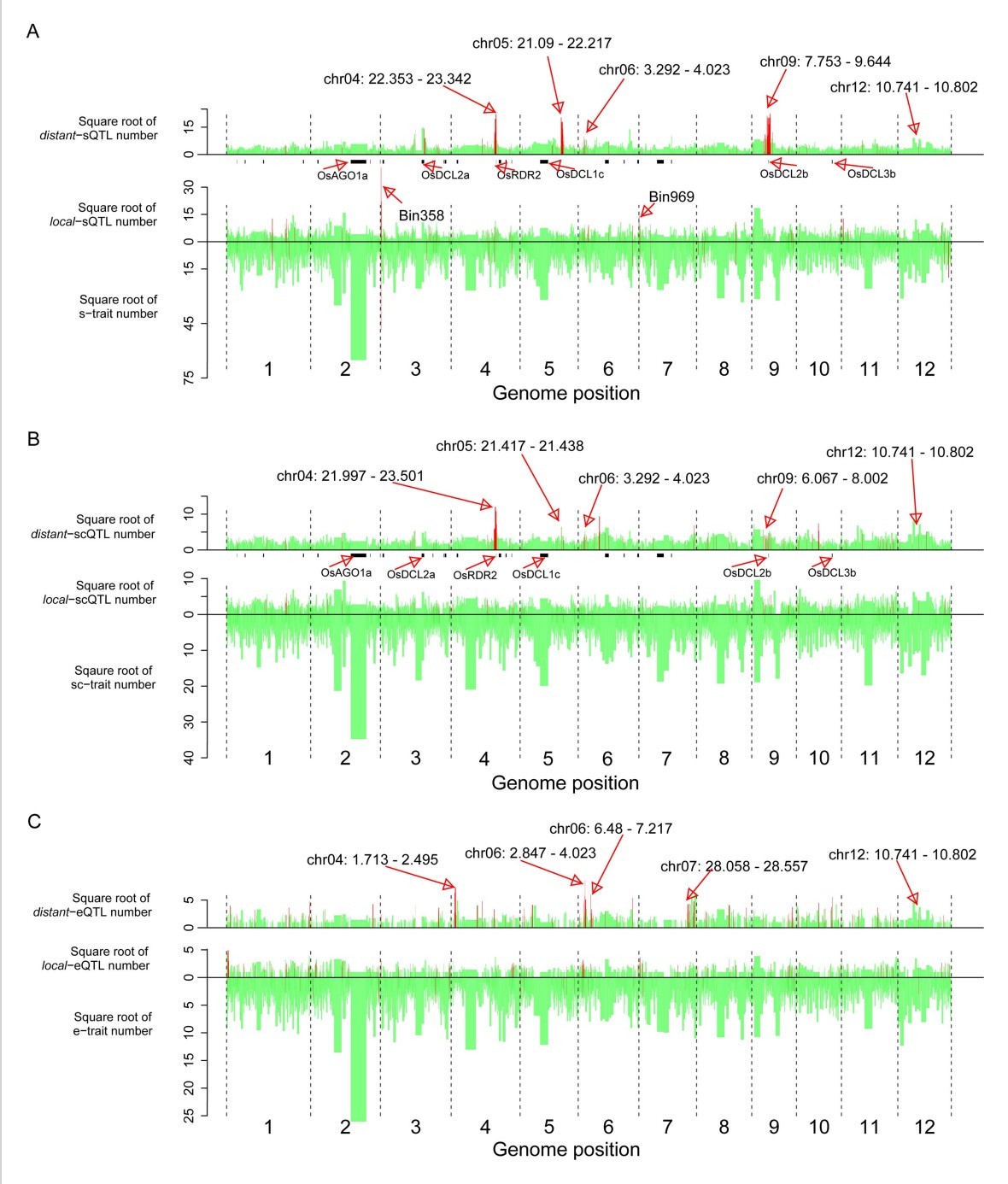

**Figure 5**. The distribution of traits and QTLs in the 1568 bins. (**A**) s-traits and sQTLs. (**B**) sc-traits and scQTLs. (**C**) e-traits and e-QTLs. The 1568 bins are arranged from left to right according to their genomic position. The width of bar represents the size of the bin. The chromosome identifiers are labeled below each plot. Bins in red color were trait or QTL hotspots. Locations for a number of interesting regions like sRNA biogenesis genes, Bin358 (LOC_Os03g01360) and Bin 969 (LOC_Os07g01240), are also indicated, in addition to the hotspots.

The following figure supplements are available for figure 5:

**Figure supplement 1**. DNA sequence polymorphisms of sRNA biogenesis genes between Zhenshan 97 and Minghui 63.

**Figure supplement 2**. Expression correlations between sRNA biogenesis genes in *distant*-sQTL hotspots and s-traits regulated by these *distant*-sQTLs.

The s-trait hotspot with the highest s-trait density was Bin969 (chr07: 0.121 Mb–0.155 Mb), holding 1025 s-traits, of which 66.6% obtained at least one sQTL (*Figure 5A*, Supplementary file 7 in Dryad [*Wang et al., 2015*]). Bin969 was also a *local*-sQTL hotspot with the third highest *local*-sQTL density. The *local*-sQTL hotspot with the highest *local*-sQTL density was Bin358 (chr03: 0.266 Mb–0.368 Mb), which held 2338 s-traits, and an overwhelming majority of these s-traits (2245) had at least one sQTL (*Figure 5A*, Supplementary file 8 in Dryad [*Wang et al., 2015*]).

## sQTL hotspots and sRNA biogenesis genes

sRNAs are usually generated by activities of Dicer-like proteins, Argonautes, and RNA-dependent RNA polymerases (RDRs) (*Kapoor et al., 2008*; *Axtell, 2013*). In rice, there are 43 transcripts of 32 genes for sRNA biogenesis including eight encoding Dicer-like proteins (*OsDCLs*), 19 Argonautes (*OsAGOs*), and 5 *OsRDRs* (Supplementary file 9 in Dryad [*Wang et al., 2015*]) (*Kapoor et al., 2008*). In this study, 26 transcripts for 20 genes were detected (FPKM ≥1) in at least one of the parents. Six transcripts of five genes (*OsDCL2a*, *OsDCL2b*, *OsAGO1d*, *OsAGO13*, and *OsAGO18*) were significantly differentially expressed between Zhenshan 97 and Minghui 63 (Supplementary file 9 in Dryad [*Wang et al., 2015*]). Comparison of DNA sequences between Zhenshan 97 and Minghui 63 revealed that 23 of the 32 sRNA biogenesis genes had polymorphisms of either SNPs or indels (*Figure 5—figure supplement 1*, Supplementary file 9 in Dryad [*Wang et al., 2015*]).

We compared the genomic positions of these sRNA biogenesis genes and *distant*-sQTL hotspots. *OsRDR2* (in Bin636 and Bin637) was located in the consecutive *distant*-sQTL hotspots (Bin632–Bin637) on chromosome 4 regulating 1416 s-traits. *OsDCL2b* (in Bin1205) was located in the consecutive *distant*-sQTL hotspots (Bin1199–Bin1210) on chromosome 9 (*Figure 5A*, Supplementary file 10 in Dryad [*Wang et al., 2015*]), which regulated the expression variation of 3759 s-traits. *OsDCL2a* was located in Bin448, which regulated the expression of 213 s-traits, although this bin was not identified as a *distant*-sQTL hotspot due to its large size (1.254 Mb). However, four neighboring bins (Bin449–Bin451, Bin453) of Bin448 were identified as *distant*-sQTL hotspots (Supplementary file 8 in Dryad [*Wang et al., 2015*]). Eight adjacent bins (Bin734–Bin741) on chromosome 5 were found to harbor 364 *distant*-sQTLs, and *OsDCL1c* was in Bin738, which contained 33 *distant*-sQTLs (Supplementary file 8 in Dryad [*Wang et al., 2015*]).

The expression correlations between all the e-traits in the *distant*-sQTL hotspots (Bin632–Bin637) on chromosome 4 and all the s-traits whose expression variations were controlled by QTLs in these bins were calculated. Most of these correlations including those between the e-traits of *OsRDR2* and all the s-traits were weak (*Figure 5—figure supplement 2A,B*). For s-traits regulated by QTLs in the consecutive *distant*-sQTL hotspots (Bin1199–Bin1210) on chromosome 9 (*Figure 5A*), most of the correlations between the s-traits and the e-traits in these bins were weak, while the correlations between s-traits and the e-traits of *OsDCL2b* were relatively strong (*Figure 5—figure supplement 2C*). About 79.7%, 17.4%, and 1.2% of the s-traits whose expression levels were regulated by the consecutive *distant*-sQTL hotspots harboring *OsRDR2* on chromosome 4 were 24 nt, 23 nt, and 22 nt, respectively. On the other hand, 51.0% of the s-traits regulated by the consecutive *distant*-sQTL hotspots harboring *OsDCL2b* on chromosome 9 were 22 nt while 35.8% were 24 nt. For s-traits regulated by *distant*-QTLs in Bin448–Bin453 (containing *OsDCL2a*), 40.8% of them were 21 nt and 24.4% were 22 nt and 24.1% were 24 nt.

Five *local*-eQTLs and six *distant*-eQTLs for nine transcripts of these sRNA biogenesis genes were identified (Supplementary file 10 in Dryad [*Wang et al., 2015*]). We also compared the genomic positions of these eQTLs with the *distant*-sQTL hotspots. A *distant*-eQTL explaining the expression variation of *OsDCL2a* was located in Bin1207, which was in the consecutive *distant*-sQTL hotspots (Bin1199–Bin1210) on chromosome 9 (Supplementary file 10 in Dryad [*Wang et al., 2015*]). The expression correlations between s-traits regulated by QTLs in these bins and the e-traits of *OsDCL2a* were mostly strong (*Figure 5—figure supplement 2D*). The functional annotations and DNA polymorphisms between the two parents for all the genes in the *distant*-sQTL hotspots harboring *OsRDR2*, *OsDCL2b*, and *OsDCL2a* were listed in Supplementary file 11 in Dryad (*Wang et al., 2015*).

The *distant*-sQTL hotspots on chromosome 5 (Bin770–Bin777) and 6 (Bin850–Bin860) were also surveyed. Approximately 98.4% and 90.5% of all the s-traits regulated by the *distant*-sQTL hotspots on chromosome 5 and chromosome 6 were 24 nt. The functional annotations of genes in the *distant*-sQTL hotspots on chromosome 5 were investigated, and no apparent relationship to the biogenesis of sRNA was found, implicating the existence of unknown mechanisms of sRNA biogenesis and/or

regulation (Supplementary file 12 in Dryad [*Wang et al., 2015*]). A gene annotated as RNA Pol II subunit Rpb7 was found in the *distant*-sQTL hotspots on chromosome 6 (Supplementary file 12 in Dryad [*Wang et al., 2015*]). Study in yeast reveals that Rpb7 has a specific role in pre-siRNA transcription (*Djupedal et al., 2005*).

## Genetic co-regulation of s-traits originating from the same mother genes

To test whether the expression of s-traits originating from the same mother gene was regulated by the same genetic mechanism, the genomic positions of QTLs for s-traits originated from the same mother gene were compared with each other. S-traits originated from LOC_Os03g01360 and LOC_Os07g01240 were not included in this calculation. We calculated correlations for the following comparisons: 97,040 pairs of s-traits that were both controlled by *local*-sQTLs (*Figure 6A*), 36,123 pairs of s-traits that were both controlled by *distant*-sQTLs (*Figure 6B*), and 51,081 pairs of s-traits in which one was controlled by *local*-sQTL and the other one by *distant*-sQTL (*Figure 6C*). For a pair of s-traits originated from the same gene controlled by *local*-sQTL, the correlation coefficients between them were mostly strong (*Figure 6A*). The correlation coefficients between s-traits controlled by the same or nearby *distant*-sQTLs were also strong (*Figure 6B*), while the correlation coefficients between s-traits controlled by *distant*-sQTLs distant from each other were relatively weak (*Figure 6B*), which represented

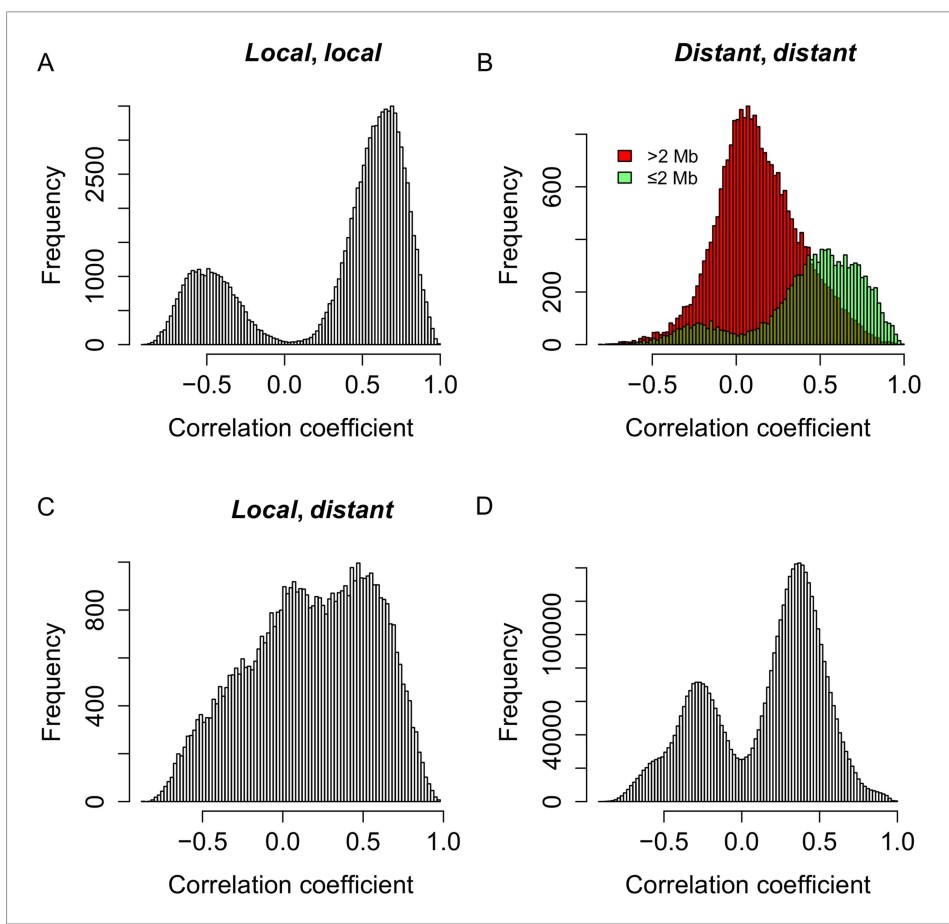

**Figure 6**. Genetic co-regulation between s-traits originating from the same mother genes. (**A**) Expression correlations between pairs of s-traits originating from the same mother genes and both regulated by *local*-sQTLs. (**B**) Expression correlations between pairs of s-traits originating from the same mother genes and both regulated by *distant*-sQTLs. Green: distance between the two *distant*-sQTLs was within two 2 Mb; red: distance between the *distant*-sQTLs was larger than 2 Mb. (**C**) Expression correlations between pairs of s-traits from the same mother genes with one regulated by *distant*-sQTL and another one regulated by *local*-sQTL. (**D**) Expression correlations between pairs of s-traits from the whole genome regulated by *distant*-QTLs located in ≤2 Mb regions.

the majority (75.0%) of the s-traits pairs. Most of the correlations between the s-traits of *local*- and *distant*-regulation were also quite high (*Figure 6C*). For comparison, we also calculated correlations for all pairs of s-traits from the entire genome not necessarily the same genes that were regulated by the same sQTLs; their expression correlations were also mostly strong, either positive or negative (*Figure 6D*), indicating that the strong correlations between the s-traits had common genetic basis.

## Genetic co-regulation of s-traits and their mother genes

To infer the commonality of the genetic controls of the expression variations for sRNAs and their mother genes, we compared the map positions of the QTLs for sRNAs and their mother genes. Again s-traits and e-traits originating from LOC_Os03g01360 and LOC_Os07g01240 were not included in this calculation. We calculated correlations for the following comparisons: 4018 s-traits and 1084 corresponding mother genes that were both controlled by *local*-QTLs (*Figure 7A*), 1721 s-traits and 695 mother genes both regulated by *distant*-QTLs (*Figure 7B*), 1840 s-traits controlled by *local*-sQTLs and 544 corresponding mother genes regulated by *distant*-eQTLs (*Figure 7C*), and 1110 s-traits and 582 mother genes controlled by *distant*-sQTLs and *local*-eQTLs, respectively (*Figure 7D*). The expression correlations between s-traits and their mother genes that were both controlled by *local*-QTLs or the same and/or nearby *distant*-QTLs were mostly strong (*Figure 7A,B*). On the other hand, the correlations between s-traits and their mother genes that were regulated by different/*distant*-QTLs were relatively weak (*Figure 7B–D*). For all the pairs of e-traits and s-traits regulated by the same QTL position with the s-traits not necessarily from the genomic region of the e-traits, their expression correlations were mostly strong (*Figure 7E*), again indicating a common genetic basis.

## The genetic effects of the sQTLs and scQTLs

There were three genotypes at each polymorphic locus in the IMF2 population: *AA*, *Aa*, and *aa*, where *A* represents the allele from one parent and *a* is the allele from the other parent. This allowed

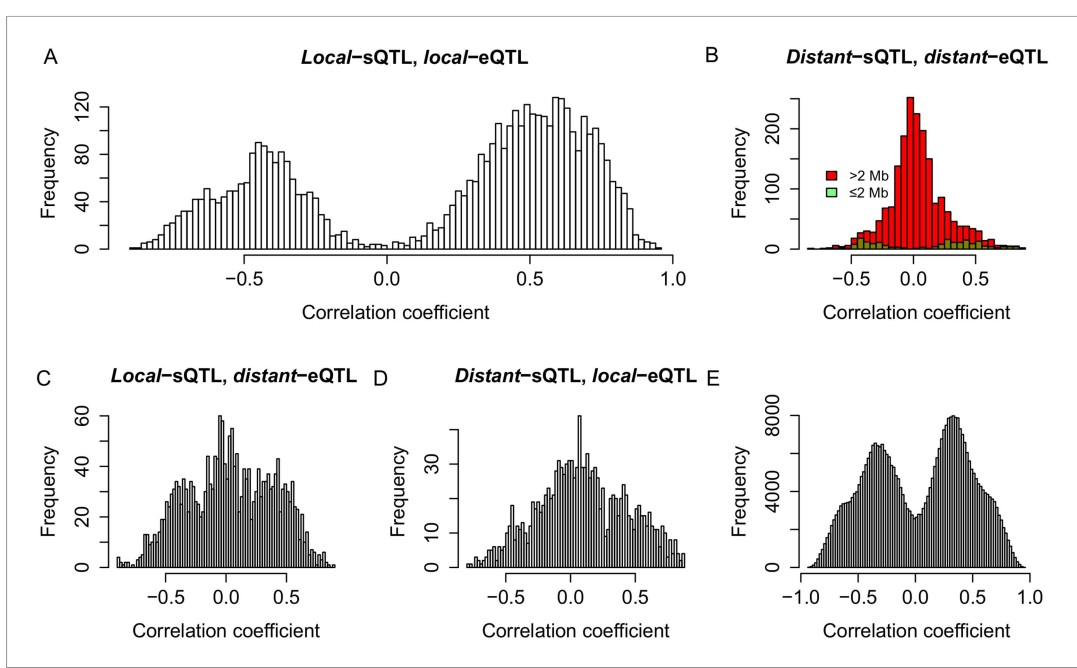

**Figure 7**. Genetic co-regulation between s-traits and their mother genes. (**A**) Expression correlations between s-traits and their mother genes that were both controlled by *local*-QTLs. (**B**) Expression correlations between s-traits regulated by *distant*-sQTLs and their mother genes also controlled by *distant*-eQTLs. Green, the distance between *distant*-QTLs for s-traits and the mother genes is within 2 Mb. Red, the distance between *distant*-QTLs is larger than 2 Mb. (**C**) Expression correlations between s-traits regulated by *local*-sQTLs and their mother genes controlled by *distant*-eQTLs. (**D**) Expression correlations between s-traits regulated by *distant*-sQTLs and their mother genes controlled by *local*-eQTLs. (**E**) Expression correlations between s-traits and e-traits from the whole genome regulated by *distant*-QTLs located in ≤2 Mb.

estimating of additive and dominant genetic effects for each of the sQTLs. Additive effect for sQTL was half of the sRNA expression difference between the two homozygotes, and dominant effect was the difference between the heterozygote and the average of the two homozygotes.

Of all the s-traits analyzed, s-traits regulated by 35,235 (49.7%) of the sQTLs exhibited higher expression levels in the Zhenshan 97 genotype than Minghui 63 genotype (Supplementary file 13 in Dryad [*Wang et al., 2015*]). We identified sQTLs with significant dominance effects using *h*-test (1000 permutations, p < 0.05) (*Huang et al., 2006*; *Zhou et al., 2012*). The analysis detected 23,018 sQTLs (32.5%) with significant dominant effects, of which the majority (71.8%) showed negative dominance such that the levels of the sRNAs were lower in heterozygotes than the mid-parent values (Supplementary file 13 in Dryad [*Wang et al., 2015*]). Moreover, 3842 (5.4%) sQTLs exhibited negative overdominance, in which sRNA levels of the heterozygotes were lower than both homozygotes. Approximately 92.4% of the sQTLs showing significant overdominance effects, either positive or negative, were *distant*-sQTLs.

The genetic effects of scQTLs were quite similar to that of sQTLs. Of all the 6913 scQTLs exhibiting significant dominant effects, 5140 (74.3%) showed negative dominance (Supplementary file 13 in Dryad [*Wang et al., 2015*]). A total of 1167 scQTLs (5.2%) exhibited negative overdominance, 89.3% of which were *distant*-scQTLs.

## A closer look of LOC_Os03g01360 and LOC_Os07g01240

We took a closer look at the two loci that contributed extremely high numbers of sRNAs. Approximately 97.4% (2278) of the s-traits produced by Bin358 were encoded by the 2-kb upstream and genic region of the locus LOC_Os03g01360, where the sRNA sequences showed extensive and consecutive overlapping (*Figure 8A*). LOC_Os03g01360 did not have homologs in *Arabidopsis* but had a homolog in maize with unknown function (http://rice.plantbiology.msu.edu/cgi-bin/ ORF_infopage.cgi?orf=LOC_Os03g01360). Three isoforms of this gene were significantly differentially expressed as measured in FPKM between Zhenshan 97 and Minghui 63 (*Figure 8—figure supplement 1A*). The sRNAs of Zhenshan 97 were mainly derived from the second and the third part of LOC_Os03g01360, while sRNAs of Minghui 63 originated from all the three parts with the expression level from the third part lower than that of sRNAs of Zhenshan 97 (*Figure 8A*). The expression levels of LOC_Os03g01360.1 and LOC_Os03g01360.4 in the genome of Minghui 63 were higher than Zhenshan 97 (*Figure 8—figure supplement 1A*). On the contrary, LOC_Os03g01360.2 was higher expressed in Zhenshan 97 than in Minghui 63. The expression of all the three transcripts in the hybrid was down-regulated compared with the mid-parent value (*Figure 8—figure supplement 1A*). We checked the cytosine methylation levels in this region, and the genomic DNA showed high cytosine methylation, especially from the start to the third exon (*Figure 8A*, 'Materials and methods'). This high methylation might be due to the enrichment of sRNAs resulting in RNA-directed DNA methylation. In the differentially expressed region of sRNAs between Zhenshan 97 and Minghui 63, such as the third exon according to the annotated gene model, Minghui 63 had lower methylation and higher mRNA transcript level than Zhenshan 97, suggesting a negative correlation between the transcript level and methylation. The expression variation of three isoforms of LOC_Os03g01360 in the IMF2 population was controlled by *local*-eQTLs (Supplementary file 7 in Dryad [*Wang et al., 2015*]). At least one sQTL each was detected for 2200 of the 2278 s-traits located in the gene LOC_Os03g01360 region, of which 2181 (84.5%) were *local*-sQTLs (Supplementary file 7 in Dryad [*Wang et al., 2015*]). The expression variation of different transcripts of LOC_Os03g01360 probably corresponded to sRNAs from specific parts of the genic region and is regulated by the same genetic factor. The majority of *distant*-sQTLs regulating the expression of s-traits originating from LOC_Os03g01360 was for 21 and 22 nt s-traits and was attributed to Bin442–Bin453 on chromosome 3.

The sRNAs in Bin969 were almost entirely derived from the intron of LOC_Os07g01240, of which 617 (61.2%) were 21-nt and 22-nt sRNAs and 280 (27.8%) were 24-nt sRNAs (*Figure 8B*). LOC_Os07g01240 had a homolog in *Arabidopsis* with unknown function and in Maize annotated as uncharacterized GPI-anchored protein (http://rice.plantbiology.msu.edu/cgi-bin/ORF_infopage.cgi? orf=LOC_Os07g01240) and was reported to modulate rice leaf rolling by regulating the formation of bulliform cells (*Xiang et al., 2012*). In all, 677 s-traits mapped 841 sQTLs, of which 514 were *local*-sQTLs (Supplementary file 7 in Dryad [*Wang et al., 2015*]). However, no significant expression difference of the two isoforms of LOC_Os07g01240 was detected between the parents

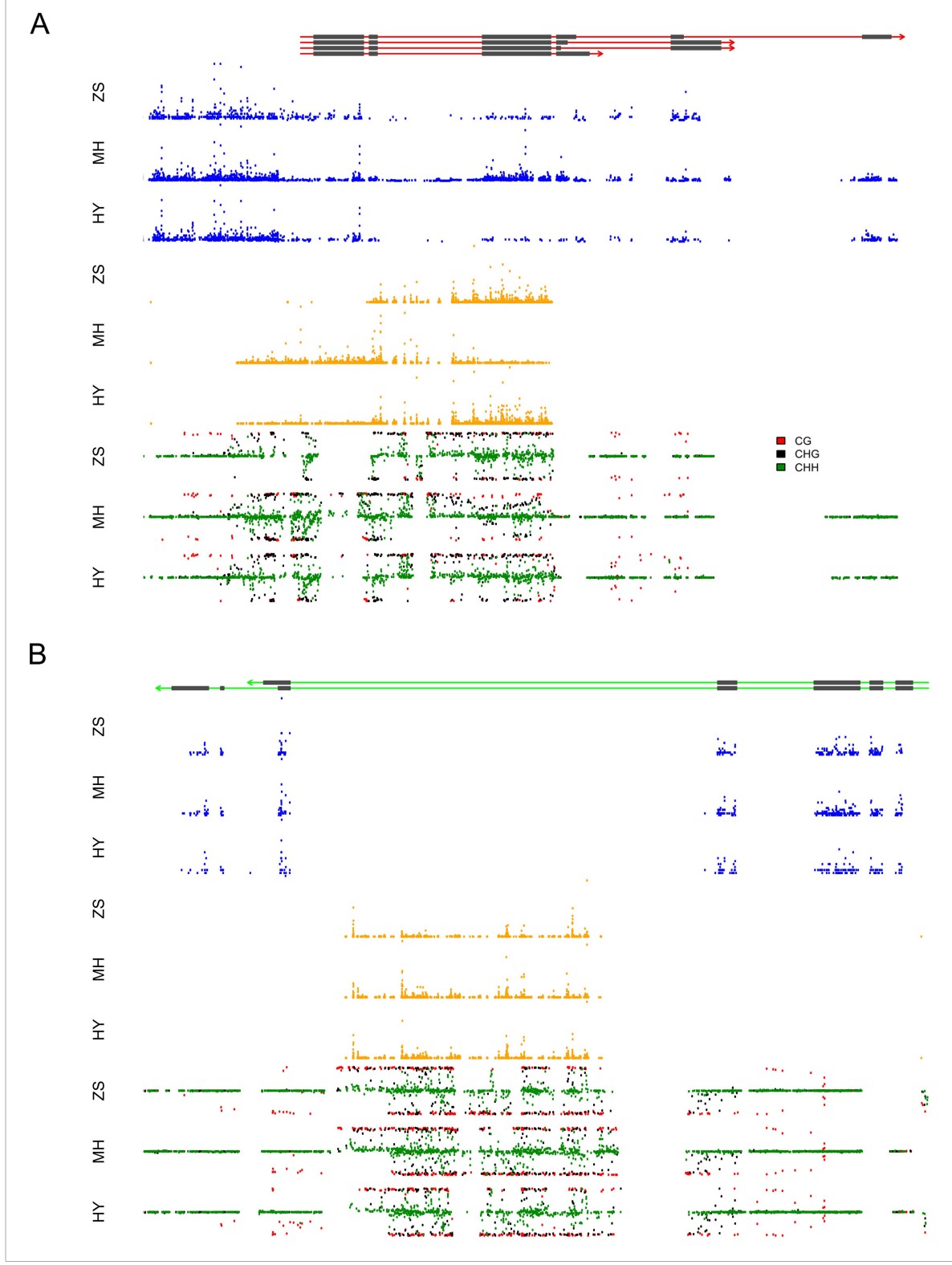

**Figure 8**. The expression levels of sRNA, mRNA, and the DNA methylation levels of the hybrid and its parents in the region of LOC_Os03g01360 and LOC_Os07g01240. (**A**) The structure of LOC_Os03g01360 is shown in the top panel, with black color representing exons and red color indicating introns and UTRs. The number of mRNA reads is shown in blue color. The sRNAs expression level is shown in orange. The DNA methylation level is displayed in

*Figure 8. Continued*

the bottom three panels. ZS, Zhenshan 97; MH, Minghui 63; HY, Hybrid. (**B**) The structure of LOC_Os07g01240 is shown in the top panel, with black color representing exons and green color indicating introns and UTRs. The meaning of different colors in the other panels is the same as in (**A**).

The following figure supplement is available for figure 8:

**Figure supplement 1**. The expression level (fragments per kb per million [FPKM]) of the three transcripts of LOC_Os03g01360 (**A**) and the two transcripts of LOC_Os07g01240 (**B**) in Zhenshan 97, Minghui 63, and the hybrid.

(*Figure 8—figure supplement 1B*), and thus no eQTL for the e-traits was detected. The majority of the 327 *distant*-sQTLs was for 21 and 22 nt s-traits and was again attributed to Bin442–Bin453 on chromosome 3. Some of the *distant*-sQTLs for 24 nt s-traits were attributed to Bin770–Bin775 on chromosome 5. The expression correlation between e-traits of LOC_Os07g01240 and s-traits originating from LOC_Os07g01240 was weak (*Figure 3B*). This result was similar to that of *Tong et al. (2013)* who showed that intronic mRNAs had a distinct expression pattern from their host genes, suggesting that intronic sRNAs might have genetic regulatory mechanisms independent of their mother genes.

## Discussion

The variation of sRNA abundance in a population depends on a range of factors: the levels of transcription of the precursors that is influenced by local elements mostly residing in the promoter regions of the sRNA itself; the mother gene or long non-coding RNAs where the sRNA is produced; polymorphisms in sequence influencing their biogenesis; and variation in distant regulatory factors located distantly from where the sRNA is generated. Our results indicated that almost one third of the detected *local*-sQTLs exhibited much larger effects than *distant*-sQTLs, consistent with the studies for eQTL of mRNA expression (*Kliebenstein, 2009*), suggesting that local elements are a major regulatory factor for sRNA variation. *Local*-QTLs regulating the expression of large numbers of traits were observed for s-traits but not for e-traits, resulting from the enrichment of s-traits in the region produced by the extensive and consecutive overlapping transcripts from the mother genes. Hotspots were found for both *distant*-QTLs of mRNA e-traits and sRNA s-traits, both of which could influence the expression variations of a large number of traits.

Our results showed that quantitative variation for a large portion of sRNAs transcribed from genic regions did not share the genetic control of the expression with the corresponding mother genes, although by common sense those sRNAs have to be transcribed along and regulated by the promoters of the mother genes. Similar results were reported in previous studies of miRNAs and their host genes, which were regulated or processed independently from their respective regulatory elements (*Siddle et al., 2014*). The results of *Monteys et al. (2010)* and *Ozsolak et al. (2008)* suggested that the transcription of sRNAs may also be regulated by their own promoters, independent of the mother genes. The widespread occurrence of such independent genetic controls between the sRNAs and the mother genes suggested that independent transcription of the sRNAs might be a common phenomenon, rather than only special circumstances. Moreover, the large number of *distant*-sQTLs that were not collocated with the *distant*-eQTLs for the corresponding mother genes supported the independent regulatory basis of the transcription of the sRNAs.

Our results showed that some of the sRNA biogenesis genes such as DCLs, AGOs, and RDRs were probably responsible for the quantitative variation of a large number of sRNAs. *OsRDR2* was found in a region of consecutive *distant*-sQTL hotspots explaining the expression variation of many sRNAs, most of which were 24 nt. This was also a region of consecutive *distant*-scQTL hotspots with the highest number of *distant*-scQTLs. *OsDCL2b* was also found in consecutive *distant*-sQTL hotspots regulating the expression of sRNAs, most of which were 22 and 24 nt. Although *OsDCL2a* and *OsDCL1c* were not in *distant*-sQTL hotspots due to lack of local recombination in that region, they were also in bins regulating the expression of a large number of s-traits. This is in accordance with the reports that RDRs function on the upstream of DCLs in the process of sRNA biogenesis (*Chapman and Carrington, 2007*). Studies in sRNA pathways in *Arabidopsis* showed that DCL2 is responsible for the synthesis of 22 nt or 24 nt siRNAs, while RDR2 functions in the production of endogenous 24 nt siRNAs and the conversion of ssRNA template into dsRNAs that serve as substrates for DCLs (*Arikit et al., 2013*), which is in good agreement with the sQTLs found in this study. However, it should also be noted that these genes were

associated with s-QTLs for only a small portions of the s-traits, while the quantitative variation of the sRNA abundance for majority of the s-traits was independent of the sRNA biogenesis genes.

One of the most interesting finding perhaps concerns the sRNAs from the two loci, LOC_Os03g01360 and LOC_Os07g01240, and their regulation patterns. Although both loci produced thousands of sRNAs, which were the most numerous in the genome, they showed sharp contrast in where the sRNAs were generated and the regulatory mechanisms with which the sRNAs were produced. sRNAs from LOC_Os03g01360 mostly originated from the 2-kb upstream and genic regions and were tightly co-regulated with each other and also with the mother gene. By contrast, sRNAs from LOC_Os07g01240 were mostly produced in the intronic region and loosely co-regulated with each other and not co-regulated with the mother gene; in fact, the s-trait variation and sQTLs were detected even without the expression variation of the mother gene. Since neither of the genes has been previously identified as related to the production and function of sRNAs, their roles in sRNA biogenesis warrant further investigation.

Another noticeable finding is the widespread negative dominance of sRNA levels detected in the IMF2 population such that heterozygotes had lower level of the sRNAs than the means of the two homozygotes (negative partial dominance) or the lower homozygote (negative overdominance). Such negative dominance was also observed in comparative sRNA profiling of hybrids relative to the parents in crosses of rice and *Arabidopsis* (*Groszmann et al., 2011b*, *2013*; *He et al., 2013*), revealing a predominant-negative regulation of siRNA expression in hybrids (*He et al., 2010*; *Chodavarapu et al., 2012*). Our genetic analysis revealed that the sQTLs showing negative dominance of the sRNA levels are mostly *distant*-sQTLs, indicating regulation by distant elements likely at the transcriptional level. Studies of transcript levels of genes (mRNAs) in the hybrid against the parents in the same rice cross also revealed more negative dominance than positive dominance, also indicating down-regulation in the hybrid relative to the parents (*Huang et al., 2006*). How the down-regulation of these two classes of transcripts was related to each other and how the two classes of down-regulation are related to the hybrid performance present great challenges for future studies.

## Materials and methods

### Plant materials and growth conditions

The plant materials consisted of an IMF2 population of 98 hybrids produced by paired crosses of 196 RILs (Supplementary file 14 in Dryad [*Wang et al., 2015*]), the two parental lines Zhenshan 97 and Minghui 63 and their hybrid. The plants were grown under normal agricultural conditions at experimental farm of Huazhong Agricultural University in the rice-growing season (May to September) in Wuhan, China. A flag leaf at the day of full expansion from each of three random plants per replicate was harvested between 17:00 and 18:00 for library construction.

### Library construction and sequencing

Total RNA was isolated from the leaf tissue using TRIzol reagent (Invitrogen, Waltham, Massachusetts), the process to convert total RNA into template suitable for high throughput DNA sequencing of mRNA-seq and sRNA-seq libraries using Sample Preparation Kit (Illumina, San Diego, California) followed the manufacturer's guide. Bisulfite sequencing (WGBS) libraries of the parents and the hybrid were made from genomic DNA isolated from the same leaf tissues as used for RNA-seq libraries. Sequencing was performed on Illumina HiSeq 2000 at BGI (http://www.genomics.cn/index, Shenzhen, China) (50 bp single end for mRNA-seq and 100 bp pair end for BS-seq).

### sRNA data processing and SNPs-replaced reference genomes of the parents

Poor-quality reads were removed using fastq_quality_filter in FASTX-Toolkit (http://hannonlab.cshl.edu/fastx_toolkit/) with parameters q = 20 and p = 85. Reads shorter than 18 nt or longer than 26 nt were excluded from further analysis. Reads mapped to rice tRNA, rRNA, snRNA, snoRNA obtained from fRNAdb (http://www.ncrna.org/frnadb/), NONCODE, GtRNAdb (http://lowelab.ucsc.edu/GtRNAdb/Osati/), and Rfam (ftp://ftp.sanger.ac.uk/pub/databases/Rfam/11.0/) were also removed.

There were 2,305,391 high-quality SNPs between Nipponbare (*Oryza sativa ssp. japonica*) and Zhenshan 97 and Minghui 63 (*Oryza sativa ssp. indica*) in the whole genome, of which 971,883 were the specific SNPs between Zhenshan 97 and Minghui 63 (data from http://211.69.128.148/rice/).

We corrected the SNP sites of the Nipponbare reference genome (http://rice.plantbiology.msu.edu, The MSU Rice Database release 7.0) according to the sequences of Zhenshan 97 and Minghui 63 to reconstruct 'SNPs-replaced reference genomes' for the two parents.

For sRNA analysis, all filtered sRNA-seq reads from Zhenshan 97 libraries were specifically mapped to the SNPs-replaced Zhenshan 97 genome and reads from Minghui 63 libraries were mapped to SNPs-replaced Minghui 63 genome. The sRNAs from heterozygous materials of $F_1$ and IMF2 were simultaneously mapped to the Zhenshan 97- and Minghui 63-replaced genomes. Bowtie (*Langmead et al., 2009*) was used to align short reads to each genome at unique genome location with no mismatch allowed. The reads from heterozygous materials could be divided into three groups: reads specifically mapped to Zhenshan 97, reads only mapped to Minghui 63 genome, and reads mapped to both Zhenshan 97 and Minghui 63 at the same position. In the third class, the sequences of the sRNAs from a monomorphic site were indistinguishable between the parents and thus obtained one count in quantitation of the abundance, whereas the two counts of the sRNAs from a polymorphic site were put together.

## Quantification of sRNA or sRNA cluster expression level

The expression level of an sRNA in a specific library was defined as the number of this sRNA divided by the total number (in millions) of genome-mapped sRNAs in this library, which was designated as 'RPM'. The R package DESeq (*Anders and Huber, 2010*) was used to quantify sRNA cluster expression level. The number of sRNA reads in each sRNA cluster for each sample was calculated and integrated as a count table, with each line representing an sRNA cluster and each column representing a sample. Then, the effective library size for each sample was estimated using the 'estimateSizeFactors' function in the DESeq package. Each column of the count table was divided by the corresponding library size to get the normalized read count, which was regarded as the expression level of the sRNA cluster.

## Processing of mRNA sequencing data

The removal of poor-quality reads for mRNA-seq reads was done in the same way as sRNA analysis. The sequences from 104 libraries were mapped to the *O. sativa ssp. japonica* (cv. Nipponbare) version 7 reference genome using TopHat (*Trapnell et al., 2009*) with default parameters. Cufflinks (*Trapnell et al., 2010*) were utilized to estimate gene expression levels according to the Nipponbare version 7 reference annotation.

## The analysis of bisulfite sequencing

For bisulfite sequencing, trimmomatic (*Lohse et al., 2012*) was used to remove low-quality reads. Bismark (*Krueger and Andrews, 2011*) was performed to align bisulfite-treated reads to the SNP-replaced genomes, allowing no mismatch in the seed of 40 nucleotides and up to two good alignments. This bisulfate mapping tool aims to find unique alignment through running four alignment processes simultaneously (*Krueger and Andrews, 2011*). Then, the de-duplication tool provided by Bismark was applied to remove potential PCR duplicates. Methylation calls were extracted for every single cytosine analyzed depending on its context (CpG, CHG, or CHH).

## QTL analysis

The ultrahigh-density bin map constructed by genotyping the RILs with population sequencing (*Xie et al., 2010*; *Yu et al., 2011*) was used. The 1568 bin genotypes of each cross in the IMF2 population were deduced from the parental genotypes (Supplementary file 6 in Dryad [*Wang et al., 2015*], *Figure 4—figure supplement 2*). CIM in R/qtl (*Haley and Knott, 1992*; *Broman and Speed, 2002*; *Manichaikul et al., 2009*) was employed to map QTLs with 1000 permutations. Additive and dominant effects were decided by 'effectscan' function in R/qtl. Variation explained by the QTL was determined using the linear QTL model as described by *Yu et al. (2011)*. The same genetic map and program parameters were used in the QTL analysis for s-traits, sc-traits, and e-traits.

## Definition of QTL and trait hotspots

The density of s-trait and sQTL was defined as the number of s-traits and sQTLs in each bin divided by the bin size (Mb), respectively. The density of sc-traits, scQTLs, e-traits, and eQTLs was calculated in

the same way. Bins with s-trait density larger than three times of the whole genome average level were defined as s-trait hotspots, while bins with sQTL density higher than six times of the whole genome average level were designated as sQTL hotspots. The process of identification of e-traits and eQTL hotspots was the same as that of s-traits and sQTLs. The definition of scQTL hotspots was identical to sQTL hotspots, while sc-traits hotspots were defined as bins with sc-traits density higher than the twice of the whole genome average.

## Additional information

### Funding

| Funder | Grant reference | Author |
|--------|-----------------|--------|
| National Natural Science Foundation of China | 31330039 | Qifa Zhang |
| The 863 Program of China | 2012AA10A304 | Qifa Zhang |

The funders had no role in study design, data collection and interpretation, or the decision to submit the work for publication.

### Author contributions

JW, Acquisition of data, Analysis and interpretation of data, Drafting or revising the article; WY, Analysis and interpretation of data, Drafting or revising the article; DZ, Acquisition of data; WX, Analysis and interpretation of data; QZ, Conception and design, Analysis and interpretation of data, Drafting or revising the article

## Additional files

### Major dataset

The following dataset was generated:

| Author(s) | Year | Dataset title | Dataset ID and/or URL | Database, license, and accessibility information |
|-----------|------|---------------|-----------------------|--------------------------------------------------|
| Wang J, Yao W, Zhu D, Xie W, Zhang Q | 2015 | Data from: Genetic basis of sRNA quantitative variation analyzed using an experimental population derived from an elite rice hybrid | http://dx.doi.org/10.5061/dryad.9d030 | Available at Dryad Digital Repository under a CC0 Public Domain Dedication. |

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
