## [Decision Letter]

Thank you for sending your work entitled “Genetic basis of sRNA quantitative variation in rice” for consideration at *eLife*. Your article has been favorably evaluated by Detlef Weigel (Senior editor) and three reviewers, including Justin Borevitz as guest Reviewing editor.

The Reviewing editor and the other reviewers discussed their comments before we reached this decision, and the Reviewing editor has assembled the following comments to help you prepare a revised submission.

You present a large body of new work, with the identification of sRNA eQTL being of wide interest. The explicit questions asked are whether sRNAs are regulated by the genes of origin, and what the genetic architecture of miRNA expression is, by making use of immortalized F2s (RILs).

The full mRNA eQTL analysis must be included. Based on this, specific tests looking for genetic co regulation of sRNA+mRNA and specific loci should be done, with independence being the null hypothesis. Also, you need to follow up the positive associations to candidate genes as suggested by other reviewers.

We would be happy to re-review a substantially revised version.

Full comments from the Reviewing editor:

[48] use an immortalized F2 (IMF2) population generated by crossing RILs from a cross between Zhenshan 97 and Minghui 63 created previously in their laboratory (23; 22) to look into the variation of small RNA expression (from RNA sequencing data). Although numerous eQTL analyses have been previously performed, the novelty is here that they look specifically to the abundance of small RNAs and that has never been done before. Moreover, the added value of using the IMF2 population in comparison to conventional RILs is that this population allows the investigation of additive and dominance effects, and heterosis. The manuscript is well written and will be comprehensible to a broad readership

The authors give a very detailed description about the number of sRNA expression traits (they refer to as e-traits), the number of cis and trans-sQTL that they find, and the overlap of the e-traits with the mRNA expression of their mother genes. The major findings are that the sRNAs are mostly independently regulated from the sRNA biogenesis genes, that the SRNA are differently transcribed from their mother genes and that most of the heterozygotes show negative dominance. Although these results on the genetic regulation of sRNAs are interesting and valuable to the plant science community, I am not yet convinced that these results have enough added value to the literature for inclusion in *eLife*.

My major point of criticism is that the analyses are very descriptive and although many sQTLs are found, the genetic regulators of sRNAs are still unknown. Furthermore, the authors might want to go into further detail for the majority of their findings.

Major points:

1) The *cis*-sQTL hotspots on chr 3 and 7 are not elaborated on and remain purely descriptive analyses. The authors could at least give information on what the function of these genes is, whether they are co-expressed with other genes, do they have homologs in Arabidopsis or other plants, etc. Trans-sQTL hotspots are also not elaborated on, except the more detailed analysis of the sRNA biogenesis genes.

2) The *OsDCL2* gene that is found to underlie a trans-sQTL and that has a *cis*-eQTL is also not analyzed further. Is there a correlation in expression between the eQTL and the sQTL? What happens if the *OsDCL2* gene is silenced or over-expressed to the sRNA expression?

3) The authors published data on yield and heterosis for the same IMF2 population previously (23; 22) and it would be very interesting to see whether the negative dominance effects are shared and correlated among the lines. Is there an effect of different sRNA expression on yield?

4) I further miss an analysis of the variation in the population, the heritability and a detailed comparison of parents and hybrid.

The other reviewer also largely agrees.

Decision recommendation:

In a population derived from crossing RIL lines, that were developed from a cross between two distinct rice inbreds (Zhenshan 97 and Minghui 63), Jian Wang and colleagues investigate: a) expression levels for small RNAs between 18 and 26 nt in length, b) mRNA expression levels and c) (genome-wide) DNA methylation by bisulfite sequencing.

Wang et al subject the small RNA expression data to an eQTL analysis and report dozens of thousands of QTLs for a similar number of e-traits.

The population consisted of 98 individuals. Hence, to accumulate the data for this project the authors produced 300 or more second generation sequencing libraries.

Zhenshan 97 and Minghui 63 are the parents of Shanyou 63, a very successful rice hybrid in China. First generation offspring of original parents (the hybrid) is heterozygous over the entire genome. But a population of individuals derived from crossing RILs provides a means to dissect the genetic interactions of this hybrid as F1s derived from different RILs will represent a different set of genes/alleles in homozygous/heterozygous states.

Hence, the chosen population is indeed suitable to elucidate genetic interactions underpinning small RNA expression; at least in this particular hybrid.

The first criticism hence concerns the title. It makes too broad a claim.

The study uses a very suitable population and a very suitable approach and technology. The study is potentially of high value for our understanding of the biology of hybrids, however, my recommendation at this point is to reject the paper. We will take another look at a revised version.

Main reasons:

I feel the study lacks a clear focus and clear messages. It is mainly a large collection of data spiced up with examples of which the reader doesn't know how representative they are.

Despite the large amount of data it is not really a resource, because the data is not accessible. Lists of QTLs is not accessible data. If a resource, then the interested reader may want to explore expression levels and methylation status himself, genome-wide, to explore the QTLs.

The low (arbitrary?) thresholds for QTL detection and arbitrary filters for including or excluding e-traits do not help in making it a resource. The reader is left with tens of thousands of QTLs of low LOD scores (see Figure 2).

I found the paper a difficult read, which was mainly due to the many different layers and concepts that one has to constantly keep separate in ones head: sRNA, siRNA, miRNA, eQTL, sQTL, e-trait, mother gene, target genes, TE genic, non-TE genic, etc., convoluted by ratios and percentages and 20(!) supplementary tables, all long list of numbers. Tens of thousands of e-traits and QTL (eQTLs? sQTLs?) described does not make it easier.

It is unclear why siRNA and miRNA are dealt with together. Given the different roles, genesis and modes of operation of the small RNAs (miRNAs and siRNA), they should be kept separate as much as possible. miRNAs are post-transcriptional control and siRNA (24mers) are part of the epigenetic machinery. It is probably not expected that they and/or their genetic control have anything in common, and the study finds ample differences. Strict separation of the 2, best in 2 different papers, would reduce confusion.

Also it is unclear why each sRNA is treated as a separate trait. Based on their biogenesis all that come from the same mother gene are jointly transcribed at least. Different abundance of sRNAs from the same ‘mother gene’ points to either interesting biology or technical artifacts. This is not explored nor discussed.

My main criticism, however, is that the presented results do not make the use of most of the data. Besides the sRNA abundance data, there is mRNA expression data (by RNAseq) for all lines, and even genome wide methylation data (by bisulfite sequencing) of all lines. These data are only mentioned in passing and selectively used in (arbitrary?) examples. If a whole genome view on sRNA control is to be presented, then the study needs to integrate over all 3 data sets (mRNA, sRNA, Methylome) in a genome wide manner.

Please address these points in a major revision focusing on highlighting the major findings supporting your sRNA regulation of mother genes. I will take a look and make a decision.

[Editors' note: further revisions were requested prior to acceptance, as described below.]

Thank you for resubmitting your work entitled “Genetic basis of sRNA quantitative variation analyzed using an experimental population derived from an elite rice hybrid” for further consideration at *eLife*. Your revised article has been favorably evaluated by Detlef Weigel (Senior editor) and a guest Reviewing editor. The manuscript has been improved but there are some remaining issues that need to be addressed before acceptance, as outlined below:

Figure 2 should be flipped along the diagonal so that QTL are along the x axis and traits on the y. This would show the vertical trans bands as is standard practice. A higher cutoff for the image or log scale coloring may highlight the major loci above the general noise.

In the Results section, *cis*-QTL may show allele specific expression. The term should be reserved for allelic tests and ‘local’ should be used when eQTL map near to their source gene.

---

## [Author Response]

*You present a large body of new work, with the identification of sRNA eQTL being of wide interest. The explicit questions asked are whether sRNAs are regulated by the genes of origin, and what the genetic architecture of miRNA expression is, by making use of immortalized F2s (RILs)*.

*The full mRNA eQTL analysis must be included. Based on this, specific tests looking for genetic co regulation of sRNA+mRNA and specific loci should be done, with independence being the null hypothesis. Also, you need to follow up the positive associations to candidate genes as suggested by other reviewers*.

Your suggestion has led to major change in the focus of the paper, which required almost complete reanalysis of the data. In the revised version, the main focus was on the co-regulation of the sRNAs, sRNAs and the mRNAs of their mother genes, mostly by correlation and QTL analysis. The manuscript now appears to be more focused. We devoted a section in the Results and a paragraph in Discussion on the two loci that produced huge numbers of sRNAs, also spent more words on the sRNA biogenesis genes. We also analyzed the possible roles of the sQTLs by relating the sQTLs especially the hotspots of *trans*-QTLs with sRNA biogenesis genes and also the length classes of the sRNAs. These changes have led to re-organizing the materials and rewriting for most of the Results section. We also made effort to unify the terminology to ease the reading.

*Major points*:

*1) The* cis*-sQTL hotspots on chr 3 and 7 are not elaborated on and remain purely descriptive analyses. The authors could at least give information on what the function of these genes is, whether they are co-expressed with other genes, do they have homologs in Arabidopsis or other plants, etc. Trans-sQTL hotspots are also not elaborated on, except the more detailed analysis of the sRNA biogenesis genes*.

We devoted a section in the Results and a paragraph in the Discussion on the two loci that produced huge numbers of sRNAs, LOC_Os03g01360 and LOC_Os07g01240 respectively. LOC_Os03g01360 does not have homolog in Arabidopsis but has homolog in Maize with unknown function (http://rice.plantbiology.msu.edu/cgi-bin/ORF_infopage.cgi?orf=LOC_Os03g01360). LOC_Os07g01240 has homologs in Arabidopsis with unknown function and in Maize annotated as “uncharacterized GPI-anchored protein” (http://rice.plantbiology.msu.edu/cgi-bin/ORF_infopage.cgi?orf=LOC_Os07g01240). This gene was reported to modulate rice leaf rolling by regulating the formation of bulliform cells (50).

We enhanced the *trans*-QTL analysis in the revised version, including the relationship the QTL hotspots with the sRNA biogenesis genes, correspondence of *trans*-sQTLs with the species (length class) of the sRNAs which made the sQTLs more meaningful.

*2) The* OsDCL2 *gene that is found to underlie a trans-sQTL and that has a* cis*-eQTL is also not analyzed further. Is there a correlation in expression between the eQTL and the sQTL? What happens if the* OsDCL2 *gene is silenced or over-expressed to the sRNA expression*?

We added the analysis to in the revised version: “*OsDCL2b* (in Bin1205) was located in the consecutive *trans*-sQTL hotspots on chromosome 9 (Figure 5, Supplementary file 10 in Dryad (47)”. For s-traits regulated by QTLs in the consecutive *trans*-sQTL hotspots (Bin1199–Bin1210) on chromosome 9, most of the correlations between them and the e-traits in these bins were weak while the correlations between them and the e-traits of *OsDCL2b* were relatively strong (Figure 5—figure supplement 2). About 51.0% of the s-traits regulated by the consecutive *trans*-sQTL hotspots harboring *OsDCL2b* on chromosome 9 were 22 nt while 35.8% were 24 nt. The *DCL2* gene of Arabidopsis was reported to be responsible for the synthesis of 22 nt or 24 nt siRNA. We have not performed the silencing or over-expression of *OsDCL2b* at present*.*

*3) The authors published data on yield and heterosis for the same IMF2 population previously (*[23]*;*
[22]*) and it would be very interesting to see whether the negative dominance effects are shared and correlated among the lines. Is there an effect of different sRNA expression on yield*?

The crosses were not the same as in the previous studies, although the RILs are. So correlations cannot be performed as suggested.

*4) I further miss an analysis of the variation in the population, the heritability and a detailed comparison of parents and hybrid*.

We added the analysis of coefficients of variation for the 165,797 s-traits (Figure 1—figure supplement 4) and selected 9 s-traits randomly from all the 165,797 s-traits to illustrate the distributions of the expression values across the IMF2 population (Figure 1—figure supplement 5). We did have data of DNA methylation, mRNA expression and sRNA expression of parents and hybrid. However, we found it hard to relate this comparison to the “genetic basis of sRNA expression” and including that part would divert the focus and make the paper unnecessarily lengthy. Since the manuscript was already criticized by one of the reviewers as having too much data, which we also agree, we decided not to include this part in the revised manuscript.

*The other reviewer also largely agrees*. *[…]*

*The first criticism hence concerns the title. It makes too broad a claim*.

We changed it to “Genetic basis of sRNA quantitative variation analyzed using an experimental population derived from an elite rice hybrid” in the revised manuscript.

*The study uses a very suitable population and a very suitable approach and technology. The study is potentially of high value for our understanding of the biology of hybrids, however, my recommendation at this point is to reject the paper. We will take another look at a revised version*.

*Main reasons*:

*I feel the study lacks a clear focus and clear messages. It is mainly a large collection of data spiced up with examples of which the reader doesn't know how representative they are*.

We had re-analyzed all the data and re-drafted the manuscript. In the revised version the main focus was on the co-regulation of the sRNAs, sRNAs and the mRNAs of their mother genes, mostly by correlation and QTL analysis. The manuscript now appears to be more focused.

*Despite the large amount of data it is not really a resource, because the data is not accessible. Lists of QTLs is not accessible data. If a resource, then the interested reader may want to explore expression levels and methylation status himself, genome-wide, to explore the QTLs*.

We made the data available in a genome browser (http://srna.ncpgr.cn/), and will submit to a public database upon acceptance of this paper.

*The low (arbitrary?) thresholds for QTL detection and arbitrary filters for including or excluding e-traits do not help in making it a resource. The reader is left with tens of thousands of QTLs of low LOD scores (see*
Figure 2*)*.

We re-analyzed the whole data. Actually, 53,613,739 unique sRNA reads were filtered to 165,797 s-traits. An sRNA with RPM value ≥0.6 was regarded as expressed, and an sRNA identified as expressed in more than 25 of the 98 IMF2s was regarded as an sRNA expression trait (s-trait). Both of the criteria are very stringent, if the huge numbers are taken into account. We performed 1,000 permutations (FDR set as 5%) for each QTL scanning and got LOD value thresholds for each QTL detected. Again we feel that the criteria are highly stringent, and the QTL scanning results in the revised manuscript were quite different from the results in the previous manuscript (see Figure 4 of the revised manuscript).

*I found the paper a difficult read, which was mainly due to the many different layers and concepts that one has to constantly keep separate in ones head: sRNA, siRNA, miRNA, eQTL, sQTL, e-trait, mother gene, target genes, TE genic, non-TE genic, etc., convoluted by ratios and percentages and 20(!) supplementary tables, all long list of numbers. Tens of thousands of e-traits and QTL (eQTLs? sQTLs?) described does not make it easier*.

Thanks for this comment. We restructured the writing by focusing on the genetics of co-regulation of the sRNAs (s-traits) and between sRNAs and the mother genes. We made efforts to unify the terminology used throughout. We deleted the part for miRNAs and condensed the writing to some extent, which reduced the text, figures and data files. We hope the paper now becomes easier to read.

*It is unclear why siRNA and miRNA are dealt with together. Given the different roles, genesis and modes of operation of the small RNAs (miRNAs and siRNA), they should be kept separate as much as possible. miRNAs are post-transcriptional control and siRNA (24mers) are part of the epigenetic machinery. It is probably not expected that they and/or their genetic control have anything in common, and the study finds ample differences. Strict separation of the 2, best in 2 different papers, would reduce confusion*.

We are grateful for this suggestion. The analysis on miRNA was removed from the revised manuscript and may be dealt with in a future paper.

*Also it is unclear why each sRNA is treated as a separate trait. Based on their biogenesis all that come from the same mother gene are jointly transcribed at least. Different abundance of sRNAs from the same ‘mother gene’ points to either interesting biology or technical artifacts. This is not explored nor discussed*.

Yes, sRNAs are jointly transcribed. However, sRNAs are not necessarily jointly transcribed from a mother gene since sRNAs were also found in the intergenic region. In this revision, we treated sRNAs transcribed from the same genomic region as an sRNA cluster, which was adopted by several previous studies (subsection headed “Expression correlations between sRNAs that originated from the same sRNA cluster”) (Shen et al. 2012). We also showed that the expression levels of sRNAs from the same mother genes or from the same clusters were slightly positively correlated (Figure 2 and Figure 2—figure supplement 5).

*My main criticism, however, is that the presented results do not make the use of most of the data. Besides the sRNA abundance data, there is mRNA expression data (by RNAseq) for all lines, and even genome wide methylation data (by bisulfite sequencing) of all lines. These data are only mentioned in passing and selectively used in (arbitrary?) examples. If a whole genome view on sRNA control is to be presented, then the study needs to integrate over all 3 data sets (mRNA, sRNA, Methylome) in a genome wide manner*.

In the revised version, we added eQTL analysis of the mRNAs, and used this for the co-regulation analysis of the sRNAs and their mother genes (subsection headed “Expression correlations between sRNAs and their mother genes”). We have data of DNA methylation for parents and hybrid, not the population, thus could not do genetic analysis.

*[Editors' note: further revisions were requested prior to acceptance, as described below*.*]*

Figure 2
*should be flipped along the diagonal so that QTL are along the x axis and traits on the y. This would show the vertical trans bands as is standard practice. A higher cutoff for the image or log scale coloring may highlight the major loci above the general noise*.

The QTL figures were already presented in the way as you suggested, the QTLs are in *x*-axis and the traits are in *y*-axis. We changed the presentation with a higher cutoff so that LOD less than 5 were not included in the presentation.

*In the Results section,* cis*-QTL may show allele specific expression. The term should be reserved for allelic tests and ‘local’ should be used when eQTL map near to their source gene*.

We changed the *cis*-QTL to *local*-QTL throughout the manuscript as suggested, which is more accurate. Accordingly, we also changed *trans*-QTL to *distant*-QTL.